# Mucosal TLR2-activating protein-based vaccination induces potent pulmonary immunity and protection against SARS-CoV-2 in mice

Current vaccines against SARS-CoV-2 substantially reduce mortality, but protection against infection is less effective. Enhancing immunity in the respiratory tract, via mucosal vaccination, may provide protection against infection and minimise viral spread. Here, we report testing of a subunit vaccine in mice, consisting of SARS-CoV-2 Spike protein with a TLR2-stimulating adjuvant (Pam$_2$Cys), delivered to mice parenterally or mucosally. Both routes of vaccination induce substantial neutralising antibody (nAb) titres, however, mucosal vaccination uniquely generates anti-Spike IgA, increases nAb in the serum and airways, and increases lung CD4[+] T-cell responses. TLR2 is expressed by respiratory epithelia and immune cells. Using TLR2 deficient chimeric mice, we determine that TLR2 expression in either compartment facilitates early innate responses to mucosal vaccination. By contrast, TLR2 on hematopoietic cells is essential for optimal lung-localised, antigen-specific responses. In K18-hACE2 mice, vaccination provides complete protection against disease and sterilising lung immunity against SARS-CoV-2, with a short-term non-specific protective effect from mucosal Pam$_2$Cys alone. These data support mucosal vaccination as a strategy to improve protection in the respiratory tract against SARS-CoV-2 and other respiratory viruses.

The emergence of SARS-CoV-2 in late 2019 and the subsequent global COVID-19 pandemic has led to substantial morbidity and mortality with currently over 600 million confirmed cases and 6.5 million deaths reported to the World Health Organisation. The development of vaccines against SARS-CoV-2 has reduced the morbidity and mortality associated with COVID-19 and limited the significant disruption to economies and freedom of movement arising from restrictive, but necessary, public health measures. Most current vaccines, either those approved for clinical use or in development, are delivered by intramuscular injection and these predominantly induce systemic neutralising Immunoglobulin G (IgG) and T-cell responses, but poor mucosal responses[1]. Serum IgG transudates into the lungs across the alveolar epithelium to reduce the risk of pulmonary viral infection, but

does not prevent infection in the nasal passages and consequent virus shedding, unless neutralising serum IgG is at very high levels[2]. Consequently, while the risk of infection and transmission is reduced by vaccination, breakthrough infections with ongoing spread can still occur[3].

Of concern is evidence for the waning of immunity post vaccination or infection[4,5], combined with the impact of viral variants of concern (VOCs), including the Omicron variant[6]. Currently, resurgence of disease in highly vaccinated populations correlates with time since vaccination and also the emergence of the Delta variant[7], for which there are suggestions that vaccine effectiveness and control of viral burden are reduced[8]. Additionally, issues surrounding vaccine equity, including affordability, transport of vaccines with complex storage

✉e-mail: richard.payne@sydney.edu; w.britton@centenary.org.au

requirements and ease of access, also remain to be fully addressed[9]. A large, partially protected population may create ideal evolutionary conditions to drive emergence of further VOCs with capacity for vaccine immune evasion characteristics or increased pathogenicity, including increased replicative capacity in the upper airway and consequent enhanced transmission. Thus, there remains a major need to continue investing in vaccine development for SARS-CoV-2.

Enhancing mucosal immunity to provide a barrier against infection may increase protection, and importantly, minimise viral spread[10]. Mucosal vaccination has shown exciting potential in pre-clinical studies for multiple respiratory pathogens, generating immunity in the lower respiratory tract to prevent morbidity and mortality. Immunoglobulin A (IgA), which is actively transported across respiratory epithelium as a stabilized dimer[11], as well as resident memory B and T cells in the lungs, nasal passages and upper airways, is elicited by intranasal vaccination and can prevent infection and reduce viral shedding[12]. In animal models, intranasal vaccination against SARS-CoV-2 can provide barrier protection against infection in both the lung and upper respiratory tract (URT), via the generation of secretory neutralizing antibodies (nAbs) and memory T-cells in the mucosa[13–19]. Currently, primarily live-attenuated viruses or viral vector-vaccines have been used for mucosal vaccination, but these may have safety concerns and cannot be used for repeat booster immunization because of anti-vector immunity. Protein vaccines overcome these concerns and provide a cost-effective strategy that is suitable in a wide range of populations, include young, aged[20] and immunocompromised groups. However protein vaccines require the inclusion of appropriate immune stimulants to enhance their immunogenicity, and effective and safe mucosal adjuvants are urgently required[10]. In addition, direct activation of the respiratory epithelium may increase the delivery of soluble proteins to gain access to antigen presenting cells (APC) for immune activation[21].

Toll-like-receptor 2 (TLR2) is an activating innate immune receptor expressed by both the respiratory epithelium and lung immune cells[22–24], and is required for effective host defence against infection[25]. We and others have recently shown that the TLR2 ligand, Pam2Cys, is a highly effective mucosal adjuvant for peptide or protein-based vaccines[26–30]. Pulmonary immunization with mycobacterial peptides or proteins conjugated to Pam2Cys stimulated protective immunity in the lung against *Mycobacterium tuberculosis* infection[26–28]. Thus, we examined the pattern of lung and systemic immune responses induced by a novel subunit vaccine, consisting of recombinant trimeric SARS-CoV-2 Spike protein admixed with the TLR2-stimulating adjuvant Pam2Cys (Pam2Cys Spike), delivered via the mucosal or systemic route. To establish the contribution of TLR2-expression by respiratory epithelia and lung immune cells to the mechanism of the mucosal vaccine response, we developed chimeric mice expressing TLR2 on haemopoietic cells or the epithelium and examined early innate responses and adaptive immunity after mucosal vaccination. Finally, the protective efficacy of the vaccine following mucosal and parenteral delivery was assessed in the K18-hACE2 murine model of SARS-CoV-2 infection[31].

## Results

### Pam2Cys Spike vaccination induces potent lung antibody and CD4+ T-cell responses

To compare immune induction by parenteral *versus* mucosal Pam2Cys Spike vaccination, C57BL/6 mice received three homologous immunizations two weeks apart, either by subcutaneous (s.c) injection or intranasal (i.n) instillation. Immunogenicity was first assessed at one-week post-final vaccination by comparison to unvaccinated mice (Fig. 1a). Vaccination by either route induced potent titres of anti-Spike IgG, including IgG1 and IgG2c isotypes in serum (up to $10^{10}$) and broncho-alveolar lavage fluid (BALF) (up to $10^7$). Compared to mice

immunized parenterally, the titres of total IgG in BALF, driven largely by IgG1, were significantly increased in mucosally immunized mice (Fig. 1b). Strikingly, anti-Spike IgA was only detected in the serum or BALF of i.n immunized mice, where it was present at very high titres (up to $10^5$). No significant anti-Spike IgE was detected in serum or BALF (Fig. S1a).

The neutralizing potential of these antibodies was assessed in a pseudovirus assay, using infection of HEK293-ACE2 cells with a SARS-CoV-2 Spike-expressing lentivirus (Fig. 1c), as well as against virulent SARS-CoV-2 (ancestral) infection of VeroE6 cells (Fig. 1d). nAb titre was defined as the dilution required to inhibit 50% of infection in either system ($EC_{50}$). Both s.c or i.n vaccination induced potent nAb titres in serum, however mucosally immunized mice had significantly greater nAb levels, and only the mucosal immunization route resulted in significantly elevated nAb in the airways.

To determine if these antibody responses were sustained, and to assess the requirement for adjuvant in enhancing antibody titre and neutralization, mice were similarly vaccinated (as in Fig. 1a) with Pam2Cys alone, Spike alone, or Pam2Cys Spike either s.c or i.n. Local and systemic immune responses were assessed at 8-weeks post final vaccination (rather than 1 week as previously). Substantial anti-Spike IgG, IgG1 and IgG2c titres (up to $10^9$) were still detected in serum from Pam2Cys Spike vaccination by either route, with a clear requirement for adjuvant to maximise antibody generation (Fig. 2a). IgA was significantly upregulated only in mucosally immunized mice, both in serum and BALF, which was dependent on adjuvant co-administration with antigen (Fig. 2b). High titres of nAb were maintained against SARS-CoV-2 (ancestral), with enhanced responses in mucosally immunized mice, particularly at the airways (Fig. 2c). Additionally, nAb in serum and BALF were assessed against Delta and Omicron (BA.5) SARS-CoV-2 variants, utilizing infection in HEK-ACE2-TMPRSS2 cells (Fig. S1b, c). While significant nAb were maintained against Delta in the serum, notably in mucosally immunized mice, this was lost against the substantially immune evasive Omicron BA.5.

The generation of local and systemic Spike-specific T-cell responses was assessed by ex vivo recall stimulation of lung and spleen leukocytes with Spike protein in the presence of Brefeldin A, intracytoplasmic cytokine staining and quantitation by flow cytometry (gating strategy in Fig. S2). At one-week post final vaccination, mucosally immunized mice had a substantial population of Spike-specific CD4+ T-cells locally in the lungs, predominantly secreting IL-17 (Fig. 3a), and a more diverse response systemically in the spleen, with IL-2, IL-17 and TNF-secreting populations significantly increased (Fig. 3b). No significantly increased populations of Spike-specific CD8+ T-cells secreting IFNγ, IL-2 or TNF were detected in response to recall with whole Spike protein (Fig. 3c, d). There was minimal inflammatory cytokine release into the airways of vaccinated mice at this time point post vaccination, with a small, but non-significant, level of IL-17A in the BALF of mucosally immunized mice (Fig. S3).

At 8 weeks post vaccination, Spike-specific CD4+ T-cells secreting IL-2, IL-17 and TNF were only maintained in the lungs of Pam2Cys Spike immunized mice (Fig. 3e), and in response to recall with a dominant CD8 epitope (Spike338-346), Spike-specific CD8+ T-cells secreting IFNγ and TNF were detected (Fig. 3f). Consistent with this was a substantial upregulation in lung T-cells with a tissue resident memory (TRM)-like phenotype, in particular CD4+ TRM (CD3+ CD4+ CD69+ CD44+ CD62L-) (Fig. 4a) that were ~25-fold more frequent than CD8+ TRM (CD3+ CD8+ CD69+ CD44+ CD62L- CD103+) (Fig. 4b). This is strongly indicative of localised cell-mediated memory responses that are generated exclusively after mucosal vaccination with Pam2Cys Spike, dependent on the presence of Pam2Cys as adjuvant. Therefore, Pam2Cys Spike vaccination was potently immunogenic, inducing powerful anti-SARS-CoV-2 nAb responses in serum, with i.n vaccination providing superior humoral and CD4+/CD8+ T-cell responses in the lungs and airways.

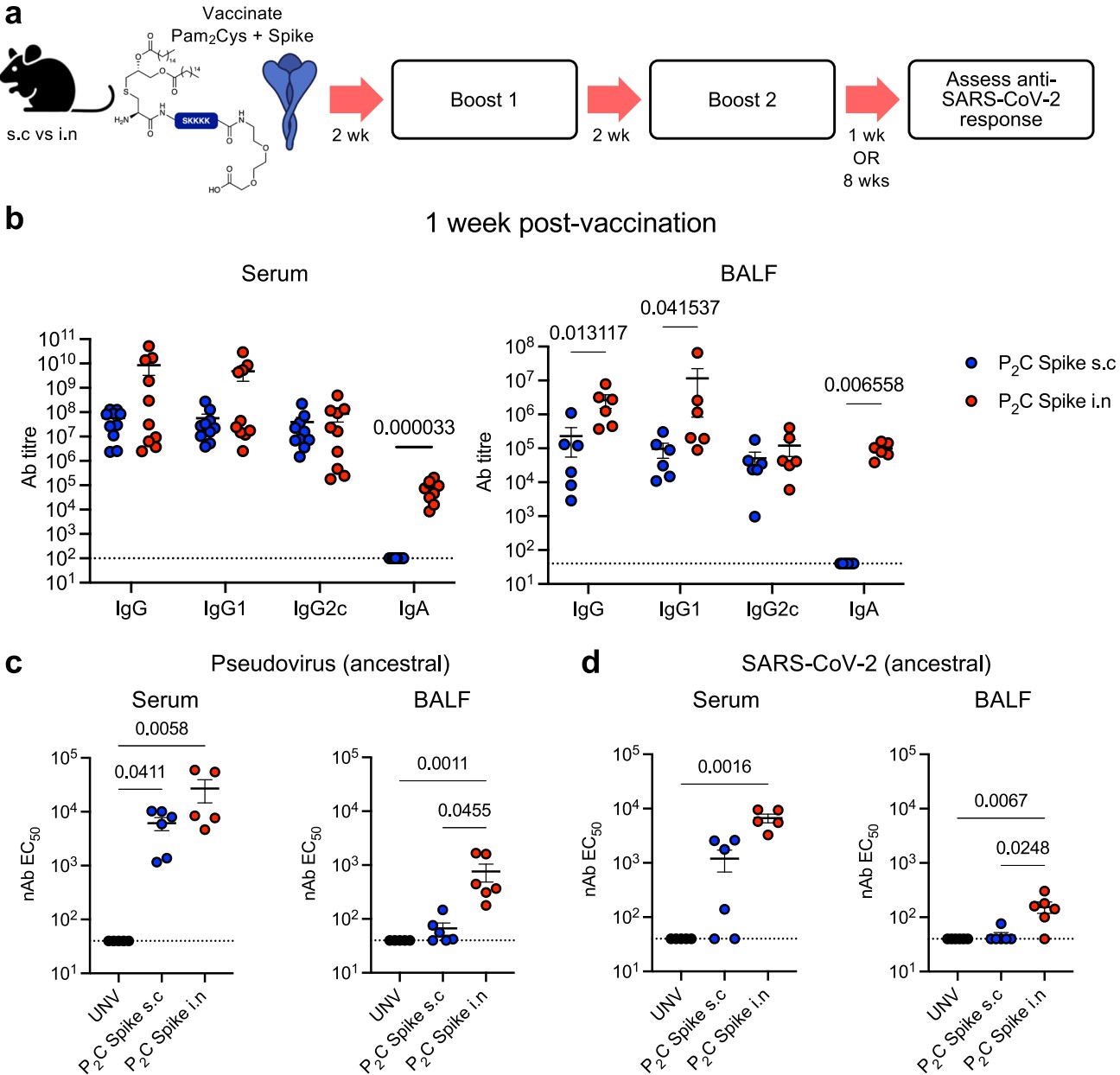

**Fig. 1 | Vaccination with Pam₂Cys Spike induces potent Spike-specific neutralizing antibody responses, which is enhanced in the airways with mucosal vaccination. a** Experimental outline. C57BL/6 mice were immunized with Pam₂Cys (P₂C) and Spike protein s.c (blue) or i.n (red) three times, two weeks (wks) apart. Immunogenicity was assessed one week (wk) post final vaccination. Control mice were unvaccinated (UNV; black). **b** Anti-spike IgG isotypes and IgA titres ($log_{10}$ scale) in the serum ($n = 10$ animals) and broncho-alveolar lavage fluid (BALF) ($n = 6$ animals), determined by ELISA, using the mean +3 SD of the level in 1:100 dilution of sera or 1:40 BALF from unvaccinated mice as the cut-off. **c** Neutralizing antibody

(nAb) titres ($log_{10}$ scale) in the serum ($n = 5$ or 6 animals) and BALF ($n = 6$ animals) of vaccinated mice were determined as the titre needed for 50% inhibition of SARS-CoV-2 Spike-expressing lentivirus in HEK293-ACE2 cells, and **d** 50% inhibition of SARS-CoV-2 infection in VeroE6 cells. Limits of detection are indicated by the dotted lines. Data points represent individual mice from two independent experiments, means +/− SEM are shown. Statistics: *p*-values indicated, (**b**) Mann–Whitney (two-sided test), (**c, d**) Kruskal–Wallis test, Dunn's multiple comparisons. Source data are provided as a Source Data file.

## TLR2 on both respiratory epithelia and lung immune cells contribute to early innate immune responses to mucosal vaccination

TLR2 is expressed by both the respiratory epithelia and lung immune cells[22–24], and is required for effective host defence against infection[25]. We hypothesised that both compartments may contribute to mucosal vaccine responses, providing an explanation for the powerful immunity induced by i.n delivery of Pam₂Cys adjuvanted vaccines. To determine the role played by each compartment, irradiated wild type (WT; BALB/c) and TLR2 deficient (*Tlr2*−/−) mice[24,25] were reconstituted with either WT or *Tlr2*−/− (KO) bone marrow (BM), to create mice with

TLR2 expression on neither compartment (KO/KO BM), immune cells only (KO/WT BM), non-immune cells only (WT/KO BM) or both compartments (WT/WT BM). Mice were rested for 12-weeks to allow hematopoietic reconstitution and replacement of radiation-resistant populations in the lungs and airways. To assess early innate responses to mucosal vaccination, mice were immunized i.n with Pam₂Cys Spike once, then immune populations of the lungs and airways examined by flow cytometry at 24-h post vaccination (Fig. 5a).

Computational analysis of lung leukocytes was first performed using the Spectre R package[32], to perform unbiased clustering of all CD45⁺ cells. To verify bone marrow replacement, relative expression of

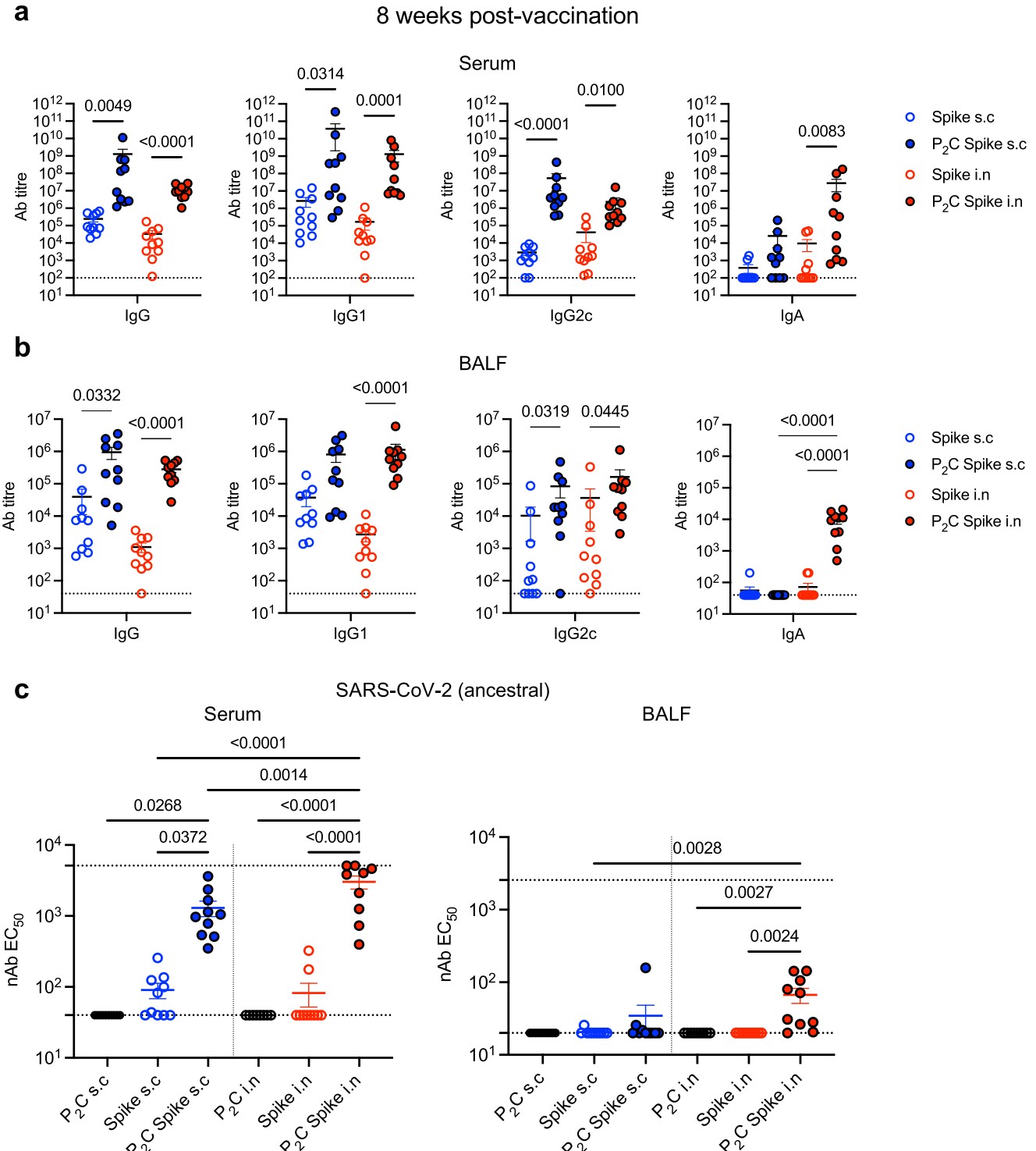

**Fig. 2 | Vaccination with Pam₂Cys Spike induces potent spike-specific neutralizing antibody responses, which is sustained post vaccination.** C57BL/6 mice were immunized with Pam₂Cys (P₂C; black) and/or Spike protein subcutaneously (s.c; blue) or intra-nasally (i.n; red) three times, two weeks apart. Immunogenicity was assessed eight weeks post final vaccination. **a** Anti-spike IgG isotypes and IgA titres in the serum and **b** broncho-alveolar lavage fluid (BALF), as determined by ELISA. **c** Neutralizing antibody titres (nAb) in the serum and BALF of vaccinated mice were determined as the titre needed for 50% inhibition of ancestral SARS-CoV-2 infection in HEK293-ACE2-TMPRSS2 cells. Upper and lower limits of detection are indicated by the dotted lines. Data are for individual mice from two independent biological experiments (*n* = 5 animals each), means +/− SEM are shown. Statistics: *p*-values indicated (**a**, **b**) Kruskal–Wallis test, Dunn's multiple comparisons, (**c**) two-way ANOVA, Tukey's multiple comparisons. Source data are provided as a Source Data file.

TLR2 was examined by Flt-SNE, with TLR2 expression above background only occurring in mice receiving WT BM (Fig. 5b, panels 2 and 4). Clusters with the highest degree of TLR2 expression were identified, based on phenotypic markers, as alveolar macrophages (SiglecF ⁺CD11cʰⁱ), neutrophils (Ly6G⁺CD11bʰⁱ), and other myeloid subsets including DC (SiglecF⁻CD11c⁺) and monocyte subsets (Ly6CʰⁱCD11bʰⁱ, Ly6CᵐⁱᵈCD11bʰⁱ) (Fig. S4). Manual gating in FlowJo was then performed to identify these TLR2-expressing subsets, as well as other immune

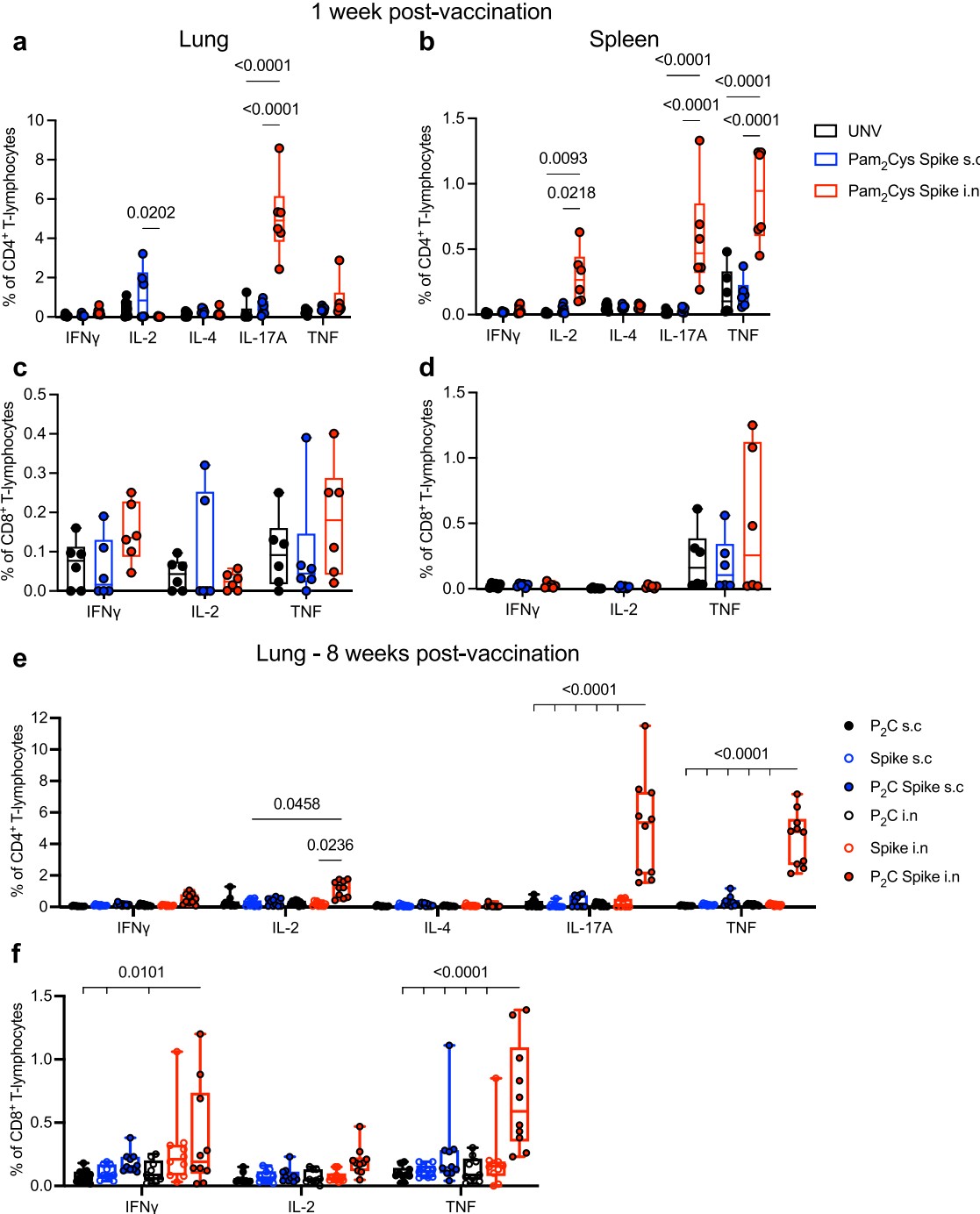

**Fig. 3 | Mucosal vaccination induces potent lung-localized and circulating IL-17-producing Spike-specific CD4+ T-cells.** C57BL/6 mice were immunized with Pam2Cys (P2C), Spike sub-cutaneously (s.c; blue) or intra-nasally (i.n; red) three times, 2 weeks apart. Control mice were unvaccinated (UNV; black **a–d**), or received P2C adjuvant only (black, **e**, **f**). The frequency of antigen-specific cytokine-producing (**a**) CD4+ T-cells in the lungs and **b** spleen, or **c** CD8+ T-cells in the lungs and **d** spleen at one week post final vaccination ($n = 6$ animals), and **e** CD4+ T-cells in the lungs and **f** CD8+ T-cells in the lungs at 8 weeks post-vaccination ($n = 10$ animals), were determined by intra-cellular staining and flow cytometry following recall stimulation ex vivo with Spike protein in the presence of Brefeldin A. Data represent individual mice from two independent biological experiments. Box and whiskers plots show median (centre), 25th and 75th percentile (box), lowest and highest value (whiskers). Statistics: $p$-values indicated, two-way ANOVA with Tukey's multiple comparisons test. Source data are provided as a Source Data file.

populations of interest (gating strategy in Fig. S5) and this confirmed that TLR2 expression on these cellular populations was only in mice receiving WT BM (Fig. S6, TLR2+ cells are indicated). A comparison of the proportion of lung immune populations revealed that mice expressing TLR2 either on non-immune or immune compartments, had a similar lung immune profile post vaccination, which was dominated by neutrophil (Ly6G+CD11bhi) influx (Fig. 5c). Quantitation of

total lung leukocytes and neutrophil count also demonstrated a consistent elevation in mice expressing TLR2 on either compartment (Fig. 5d). Notably, however, innate cells that include classic antigen presenting cells, including dendritic cells (DC; SiglecF−CD3−B220−CD11c+MHCIIhi), alveolar macrophages (SiglecF+CD11chi) and a monocyte subset (SiglecF−CD3−B220−CD11c−CD11bhiLy6Cmid), were significantly elevated in mice that had TLR2 expression on non-immune cells, or

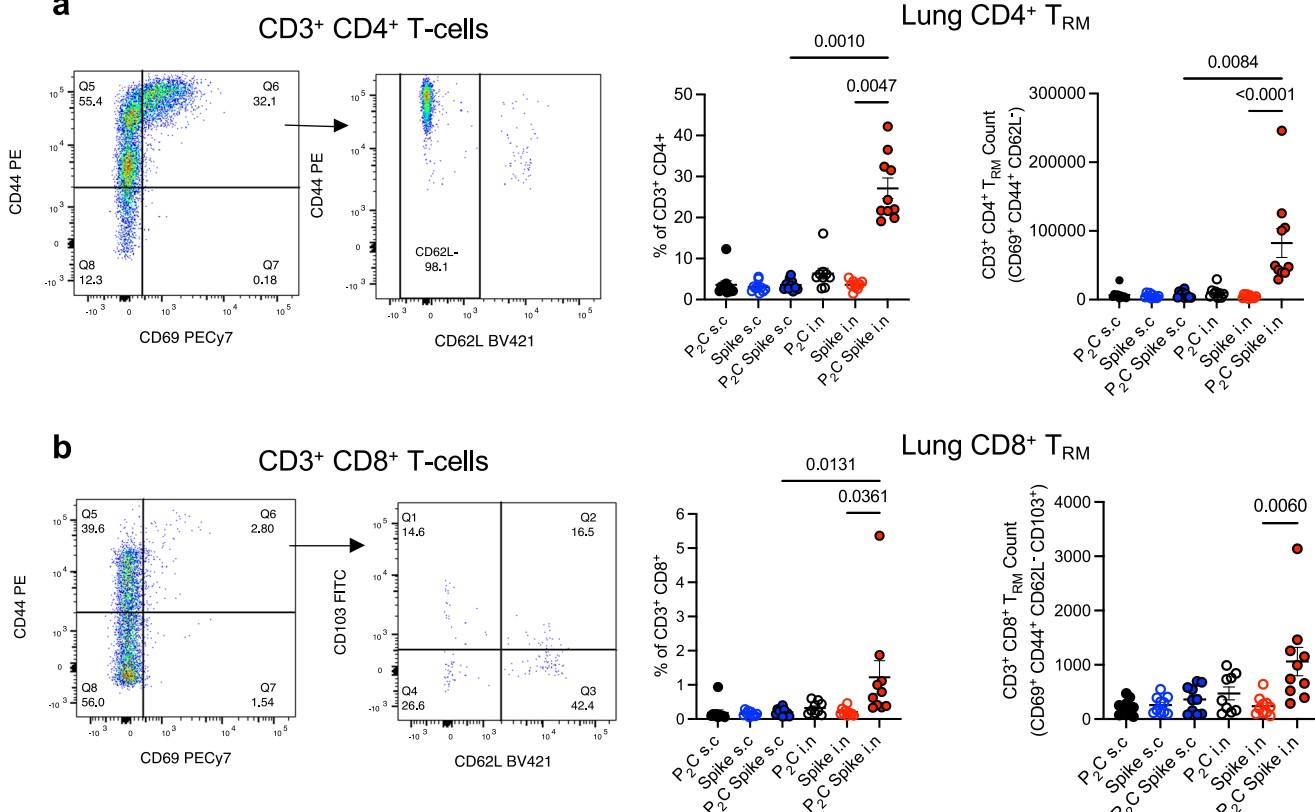

**Fig. 4 | Mucosal vaccination induces lung localized Tissue Resident Memory (T$_{RM}$)-like T-cells.** C57BL/6 mice were immunized with Pam$_2$Cys (P$_2$C; black) and/or Spike sub-cutaneously (s.c; blue) or intra-nasally (i.n; red) three times, 2 weeks apart. At 8 weeks post-final vaccination (**a**) T$_{RM}$-like CD4$^+$ T-cells (defined as CD3$^+$ CD4$^+$ CD69$^+$ CD44$^+$ CD62L$^-$) and **b** T$_{RM}$-like CD8$^+$ T-cells (defined as CD3$^+$ CD8$^+$ CD69$^+$ CD44$^+$ CD62L$^-$ CD103$^+$) in the lungs were identified by flow cytometry. Gating strategy, frequency and enumeration are shown. Data are for individual mice from two independent biological experiments ($n = 5$ each), means +/− SEM are shown. Statistics: $p$-values indicated, Kruskal−Wallis, Dunn's multiple comparisons. Source data are provided as a Source Data file.

both non-immune and immune cells (Fig. 5d). This was despite the expression of TLR2 on these cells only in mice that received WT BM (Fig. S6). Thus, activation of TLR2 on the respiratory epithelia plays a key role in recruiting these innate immune cells to the lungs early after mucosal vaccination.

Examination of immune phenotypes in BALF samples revealed a similar pattern, whereby mice expressing TLR2 on either immune or non-immune cells had a similar profile of immune recruitment into the airways, dominated by neutrophils (Fig. S7a). Quantitation of total BALF leukocytes and neutrophils also demonstrated a consistent elevation in mice expressing TLR2 on either compartment (Fig. S7b). By contrast, other innate populations displayed a different pattern in BALF compared to lung tissue. Minimal changes were observed in alveolar macrophages and DCs, but elevation of a monocyte subset (SiglecF$^-$CD3$^-$B220$^-$CD11c$^-$CD11b$^{hi}$Ly6C$^{mid}$) occurred in all groups expressing TLR2, and a significant elevation of inflammatory monocytes (SiglecF$^-$CD3$^-$B220$^-$CD11c$^-$CD11b$^{hi}$Ly6C$^{hi}$) was observed only in mice expressing TLR2 on immune cells (Fig. S7b). TLR2 expression on these cells was confirmed to be only in groups of mice receiving WT BM (Fig. S8, TLR2$^+$ cells are indicated). TLR2 expression by hematopoietic cell lineages led to early cytokine responses in the airways at 24 h after mucosal vaccination with Pam$_2$Cys Spike, notably TNF and IL-6, consistent with alveolar macrophage and monocyte activation (Fig. S9). Thus, TLR2 on both respiratory epithelia and immune cells plays a role in early innate immune responses to Pam$_2$Cys Spike mucosal vaccination. Expression on either compartment is sufficient for neutrophil recruitment, but expression on non-immune cells leads to increased populations of antigen presenting cells, such as DC, in the lungs.

**TLR2 on immune cells is required for optimal adaptive responses to mucosal Pam$_2$Cys Spike vaccination**

To determine the impact of TLR2 expression on non-immune *versus* immune cells on the generation of adaptive immune responses, a similar experimental protocol was performed, but mice received three i.n vaccinations with Pam$_2$Cys Spike, each 2 weeks apart (Fig. 6a). The immune profile in the lungs was examined 1 week post final booster vaccination. Computational clustering analysis of all lung leukocytes was again performed to examine the relative expression of TLR2 and verify bone marrow replacement (Fig. 6b). At this time point after booster vaccination, clusters with the highest degree of TLR2 expression were identified as alveolar (SiglecF$^+$CD11c$^{hi}$) and interstitial (SiglecF$^-$CD64$^+$CD11b$^{hi}$) macrophages (Fig. S10). Manual gating in FlowJo was performed to quantitate these cells and confirm their TLR2 expression, and examine other immune cell populations of interest (Fig. S11). There was a significant increase in total lung leukocytes in groups expressing TLR2 on immune cells, and of interest, a substantial increase in both the proportion and number of T-cells, particularly CD4$^+$ T-cells (Fig. 6c, d). A significant increase of B-cells (B220$^+$CD11c$^-$) was apparent in *Tlr2−/−* mice receiving WT BM, but not in WT mice receiving WT BM. In contrast, mice without TLR2 on immune cells against either a WT or *Tlr2−/−* background, had a similar profile of lymphocyte populations.

Spike antigen-specific humoral and T-cell responses were also examined. While all mice had detectable Spike-specific IgG responses in serum and BALF, expression of TLR2 on either immune or non-immune cells was required for the generation of substantial titres of IgG or IgA (Fig. 7a). There was a clear advantage in mice with

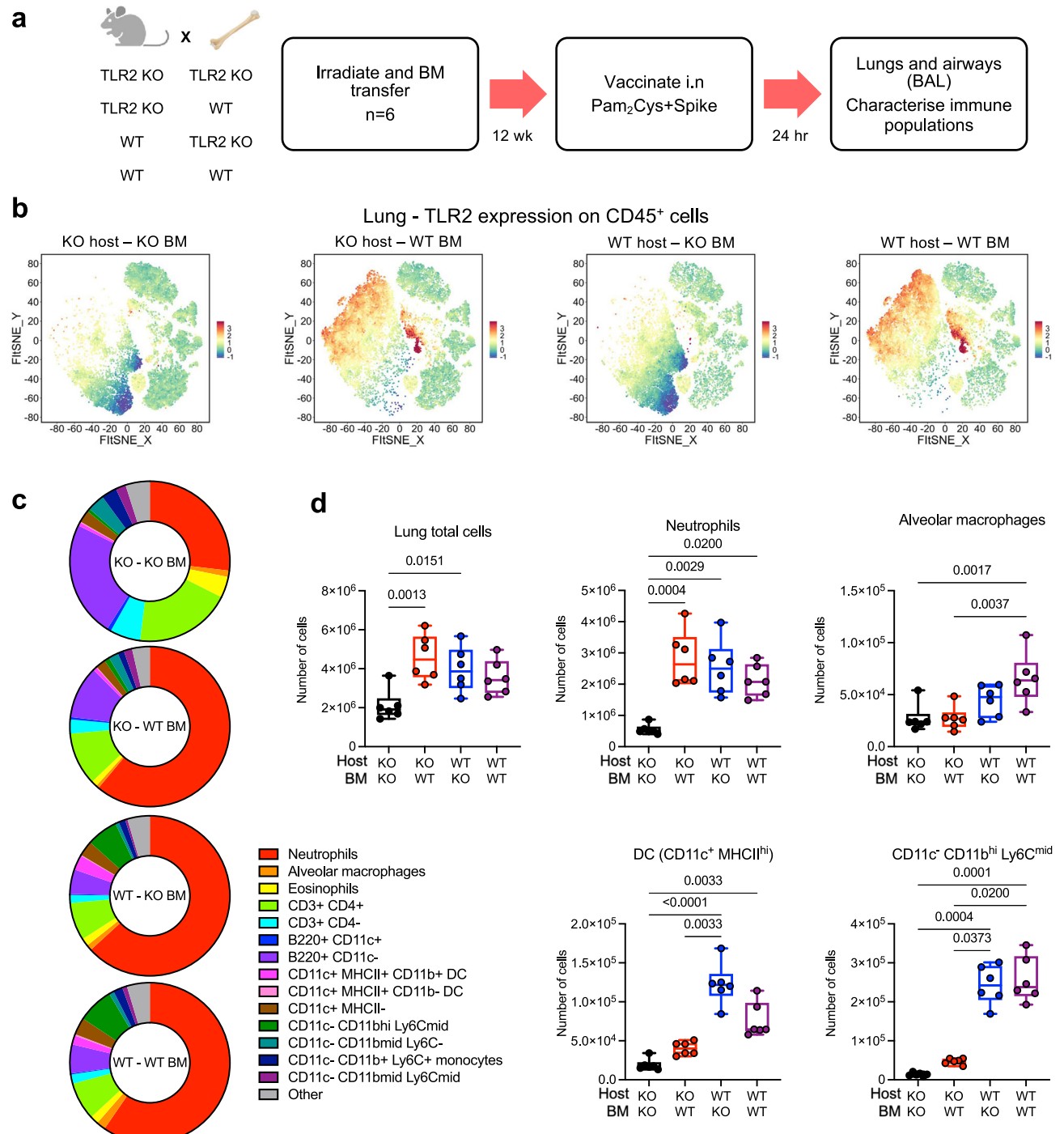

**Fig. 5 | TLR2 expression by both hematopoietic and non-hematopoietic lineages contribute to early innate immune responses to mucosal vaccination with Pam₂Cys Spike. a** Experimental outline. *Tlr2−/−* (KO) or wild type (WT) mice (*n* = 6) were irradiated then received transfer of *Tlr2−/−* or WT bone marrow (BM) cells intravenously (i.v.). Mice were rested for 12 weeks to allow hematopoietic reconstitution and replacement, then immunized once with Pam₂Cys Spike i.n. Immune cells in the lungs were characterized at 24 h post vaccination. **b** To verify bone marrow replacement, unsupervised clustering of lung CD45⁺ cells was performed on flow cytometry data and relative expression of TLR2 was examined by Flt-SNE for each experimental group. **c** Mean proportions of CD45⁺ cell populations in the lungs were determined by flow cytometry, with (**d**) quantitation of populations of interest (KO host KO BM, black; KO host WT BM, red; WT host KO BM, blue; WT host WT BM, purple). Data are for individual mice (*n* = 6). Box and whiskers plots show median (centre), 25th and 75th percentile (box), lowest and highest value (whiskers). Statistics: *p*-values indicated, Kruskal−Wallis test, Dunn's multiple comparisons. Source data are provided as a Source Data file.

expression of TLR2 on immune cells, with these groups generating significantly greater IgG and IgA titres, and importantly adjuvant signalling via TLR2 was required to generate nAb in both serum and BALF (Fig. 7b). Consistent with this finding, while there was no change between groups in the proportion of total CD3⁺CD4⁺ cells in the mediastinal lymph node (MLN), mice with TLR2 on immune cells had a significant increase in the proportion of Bcl6⁺CD4⁺ T-cells, consistent with an increased T-follicular helper (Tfh) response (Fig. 7c, gating strategy in Fig. S12). In the lung tissue, Spike-specific cytokine secreting CD4⁺ T-cells again were significantly upregulated only in groups with TLR2 on immune cells, in particular IL-17A⁺ and TNF⁺ cells (Fig. 7d). Systemically in the spleen, Spike-specific cytokine-secreting CD4⁺

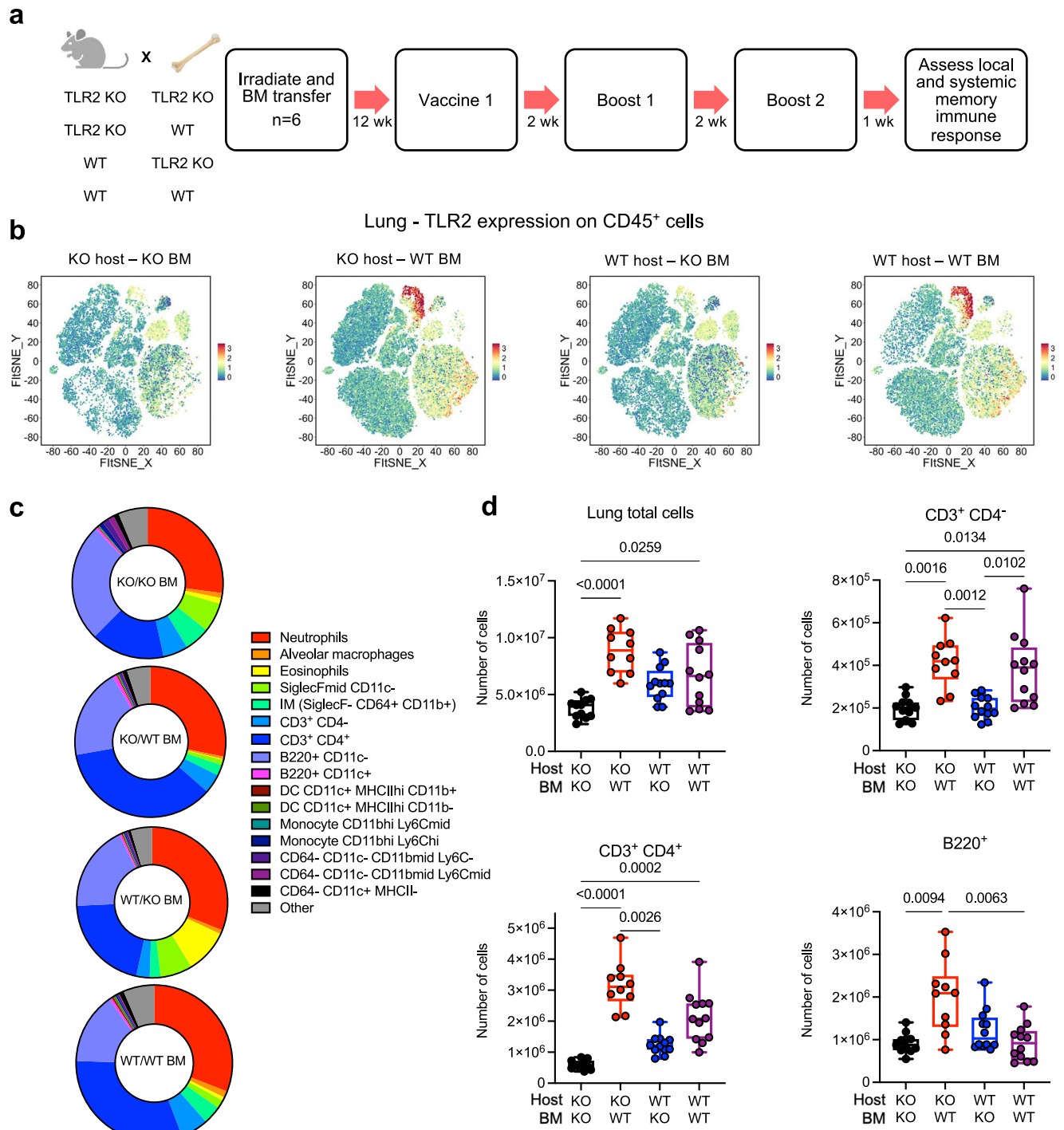

**Fig. 6 | TLR2 expression by the hematopoietic lineage is required for optimal T-cell responses in the lung following mucosal vaccination and boosting with Pam₂Cys Spike. a** Experimental outline (week; wk). *Tlr2⁻/⁻* (KO) or wild type (WT) mice were irradiated then received transfer of *Tlr2⁻/⁻* or WT bone marrow (BM) cells i.v. Mice were rested for 12 weeks to allow hematopoietic reconstitution and replacement, then immunized with Pam₂Cys Spike intra-nasally (i.n) three times, two weeks apart. Immune responses in the lungs were characterized at one week post final vaccination. **b** To verify bone marrow replacement, unsupervised clustering of lung CD45⁺ cells was performed on flow cytometry data and relative

expression of TLR2 was examined by Flt-SNE for each experimental group. **c** Mean (*n* = 6) proportions of CD45⁺ cell populations in the lungs determined by flow cytometry, representative of two independent biological experiments, with (**d**) quantitation of populations of interest (KO host KO BM, black; KO host WT BM, red; WT host KO BM, blue; WT host WT BM, purple). Data are for individual mice from two independent biological experiments (*n* = 6 animals in each). Box and whiskers plots show median (centre), 25th and 75th percentile (box), lowest and highest value (whiskers). Statistics: *p*-values indicated, Kruskal−Wallis test, Dunn's multiple comparisons. Source data are provided as a Source Data file.

T-cells were detectable in all groups that expressed TLR2 on either non-immune or immune cells, but again, TLR2 expression on immune cells provided an advantage, and expression on both compartments provided the greatest response (Fig. 7e). Thus, TLR2 expression is

required for the generation of consequential adaptive immune responses to the Spike protein following mucosal Pam₂Cys Spike vaccination, and expression of TLR2 on immune cells is required for optimal generation of adaptive immunity.

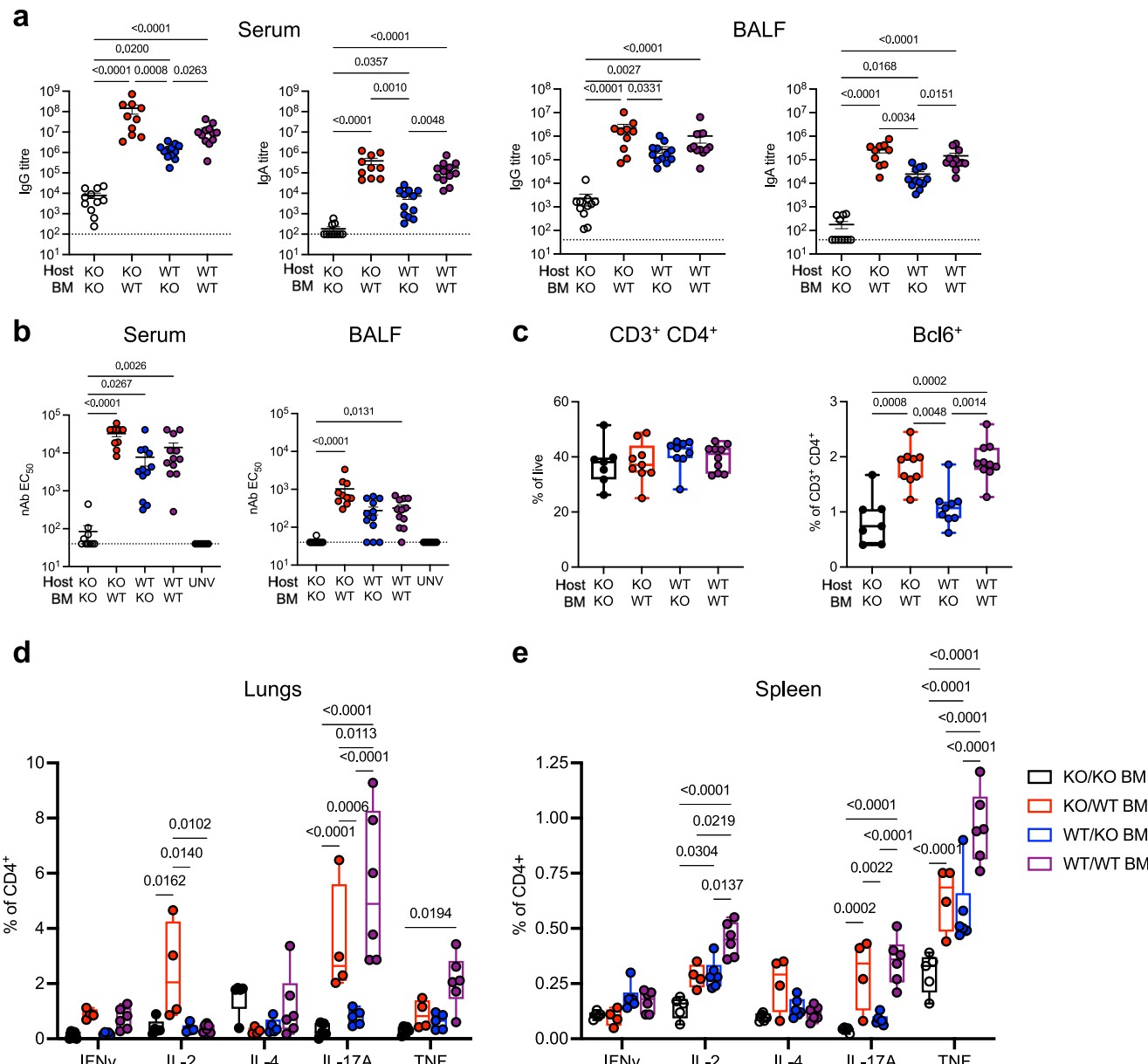

**Fig. 7 | TLR2 expression by the hematopoietic lineage is required for optimal Spike-specific antibody and CD4+ T-cell responses in the lung following mucosal vaccination and boosting with Pam2Cys Spike.** *Tlr2−/−* (KO) or wild type (WT) mice were lethally irradiated then received transfer of *Tlr2−/−* or WT bone marrow (BM) cells i.v. Mice were rested for 12 weeks to allow hematopoietic reconstitution and replacement, then immunized with Pam$_2$Cys Spike i.n three times, two weeks apart. Immune responses were characterized at one week post final vaccination. Control mice were unvaccinated *Tlr2−/−* (UNV). **a** Anti-spike IgG and IgA titres (log$_{10}$ scale) in serum and bronchoalveolar lavage fluid (BALF) determined by ELISA. **b** Neutralizing antibody (nAb) titres (log$_{10}$ scale) in serum and BALF of vaccinated mice were determined as the titre needed for 50% inhibition of SARS-CoV-2 Spike-expressing lentivirus in HEK293-ACE2 cells. Limits of detection

are indicated by the dotted lines. **c** Proportion of CD4+ T-cells and Bcl6+ CD4+ T-cells in mediastinal lymph nodes. Frequency of antigen-specific cytokine-producing CD4+ T-cells in the **d** lungs and **e** spleen, detected by intra-cellular staining and flow cytometry following recall with Spike protein in the presence of Brefeldin A. KO host KO BM, black; KO host WT BM, red; WT host KO BM, blue; WT host WT BM, purple. Statistics: *p*-values indicated, means +/− SEM are shown. Box and whiskers plots show median (centre), 25th and 75th percentile (box), lowest and highest value (whiskers). **a**–**c** Data are for individual mice from two independent biological experiments (*n* = 6 each), Kruskal–Wallis test, Dunn's multiple comparisons. **d**, **e** Representative of two independent biological experiments (*n* = 6 each), two-way ANOVA, Tukey's multiple comparisons test. Source data are provided as a Source Data file.

## Pam$_2$Cys Spike vaccination induces sterilizing immunity against SARS-CoV-2 challenge in mice

To determine if the potent immune responses induced by Pam$_2$Cys Spike vaccination were protective, an in vivo murine model of SARS-CoV-2 infection was utilized. C57BL/6 mice expressing human ACE2 (K18-hACE2) were immunized with Pam$_2$Cys adjuvant alone, or Pam$_2$-Cys Spike either s.c or i.n, three times two weeks apart. Serum was collected at one week post final vaccination, and the induction of

anti-Spike IgG and IgA responses were confirmed. As in wild type C57BL/6 mice (Fig. 1b), mucosal vaccination induced superior titres and uniquely generated IgA (Fig. 8a). The neutralizing potential of these antibodies to inhibit viral infection and cytopathic effect, was first tested in vitro in HEK293 cells overexpressing ACE2 and TMRPSS2 (HEK-ACE2-TMRPSS2), utilizing the ancestral (A.2.2) SARS-CoV-2 strain, and also Beta and Delta variants. Substantial nAb titres were generated against the ancestral strain, and while these titres were

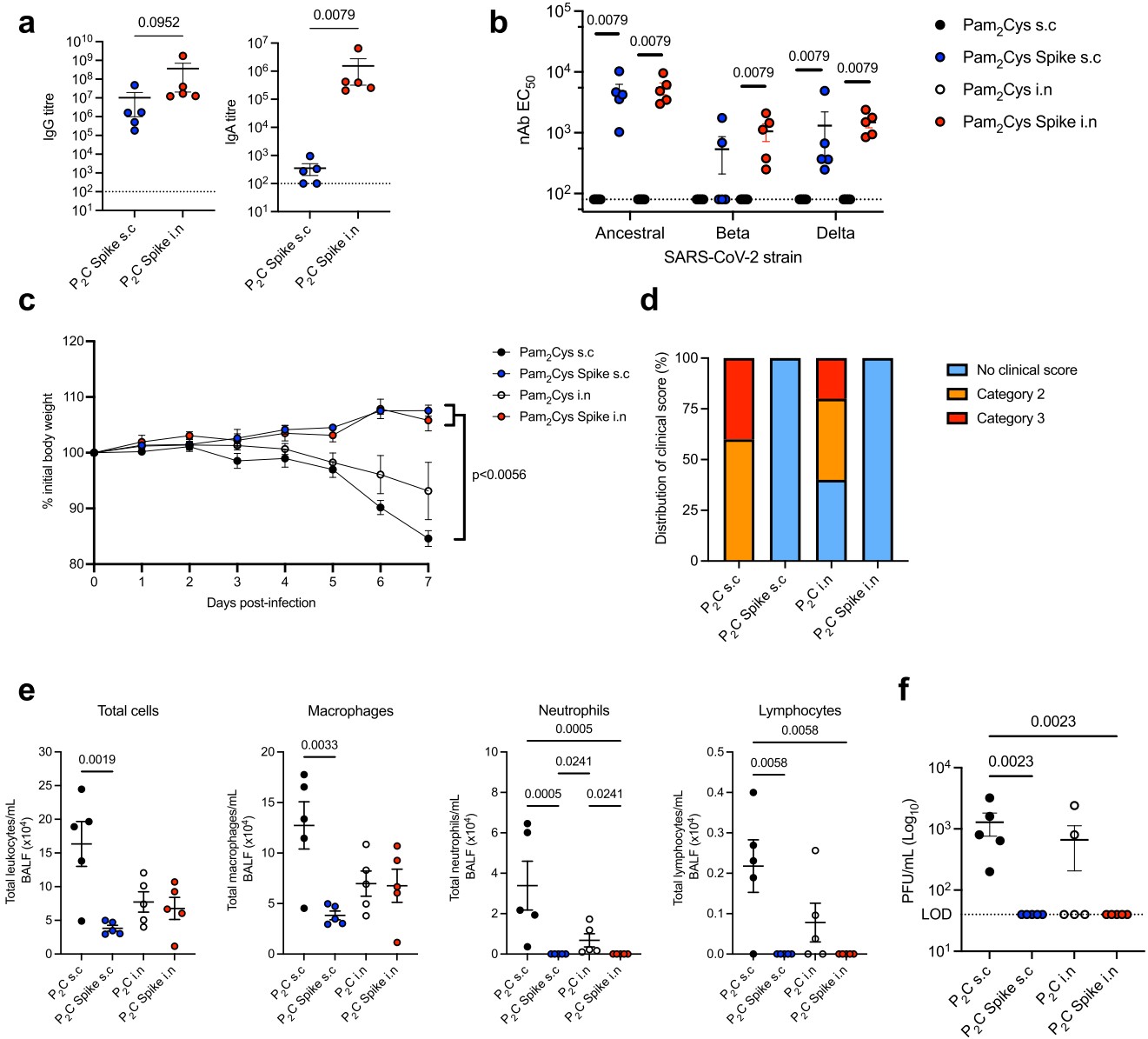

**Fig. 8 | Vaccination with Pam$_2$Cys Spike induces potent protection against SARS-CoV-2 challenge.** K18-hACE2 mice ($n = 5$) were immunized with Pam$_2$Cys alone (P$_2$C; black), or Pam$_2$Cys Spike, sub-cutaneously (s.c; blue) or intra-nasally (i.n; red) three times, two weeks apart. Three weeks post final vaccination mice were challenged with $10^3$ plaque forming units (PFU) of SARS-CoV-2 i.n and protective efficacy assessed. **a** Anti-Spike IgG and IgA titres (log$_{10}$ scale) in serum collected one week post final vaccination, determined by ELISA, and **b** neutralizing antibody (nAb) titres (log$_{10}$ scale) were determined as the titre needed for 50% inhibition of SARS-CoV-2 infection (ancestral, Beta or Delta variant) in HEK293-ACE2-TMPRSS2 cells. **c** Mouse weight loss from days 0–7 post infection (p.i) with SARS-CoV-2. **d** Clinical score distribution, increasing category indicates severity, (no clinical score, blue; category 2, orange; category 3, red) (**e**) total leukocyte and differential cell counts in the BALF and **f** lung viral load (PFU) at day 7 p.i. Limits of detection (LOD) indicated by the dotted lines. Data are for individual mice ($n = 5$ each), means +/− SEM are shown. Statistics: *p*-values indicated. **a, b** Mann–Whitney test, (**c**) Two-way ANOVA with Tukey's multiple comparisons, (**e, f**) Kruskal–Wallis test, Dunn's multiple comparisons. Source data are provided as a Source Data file.

reduced against Beta and Delta strains, mucosal immunization induced significant and superior responses, with a mean nAb EC$_{50}$ greater than $10^3$ against both VOCs (Fig. 8b).

At 3 weeks post final booster vaccination, mice were challenged with SARS-CoV-2 i.n, and clinical data collected for 7 days. Mice receiving Pam$_2$Cys Spike vaccination, by either route, were completely protected from weight loss (Fig. 8c) and clinical indicators of disease (Fig. 8d). In contrast, mice immunized s.c with Pam$_2$Cys adjuvant only had no protection, however, 40% of mice receiving Pam$_2$Cys i.n notably did not lose weight or show clinical indicators of disease. A substantial influx of leukocytes in BALF was observed in Pam$_2$Cys s.c mice, including macrophages, neutrophils and lymphocytes, which were

significantly reduced in the Pam$_2$Cys Spike s.c group. Interestingly, total leukocyte counts were reduced equivalently in both Pam$_2$Cys i.n and Pam$_2$Cys Spike i.n vaccinated mice, although neutrophils were significantly elevated in Pam$_2$Cys i.n mice (Fig. 8e). Consistent with the substantial antibody response generated by Pam$_2$Cys Spike, vaccination by either route induced sterilizing protection with no virus detected in the lung tissue at day 7 post infection (Fig. 8f). Intriguingly, 60% of mice receiving Pam$_2$Cys i.n also had no detectable virus in lung tissue. Consistent with these data, mice vaccinated with Pam$_2$Cys Spike by either route showed a marked reduction in inflammatory cell infiltration in the lung parenchyma observed by histological examination (Fig. 9). As previously observed, mice that received Pam$_2$Cys i.n had

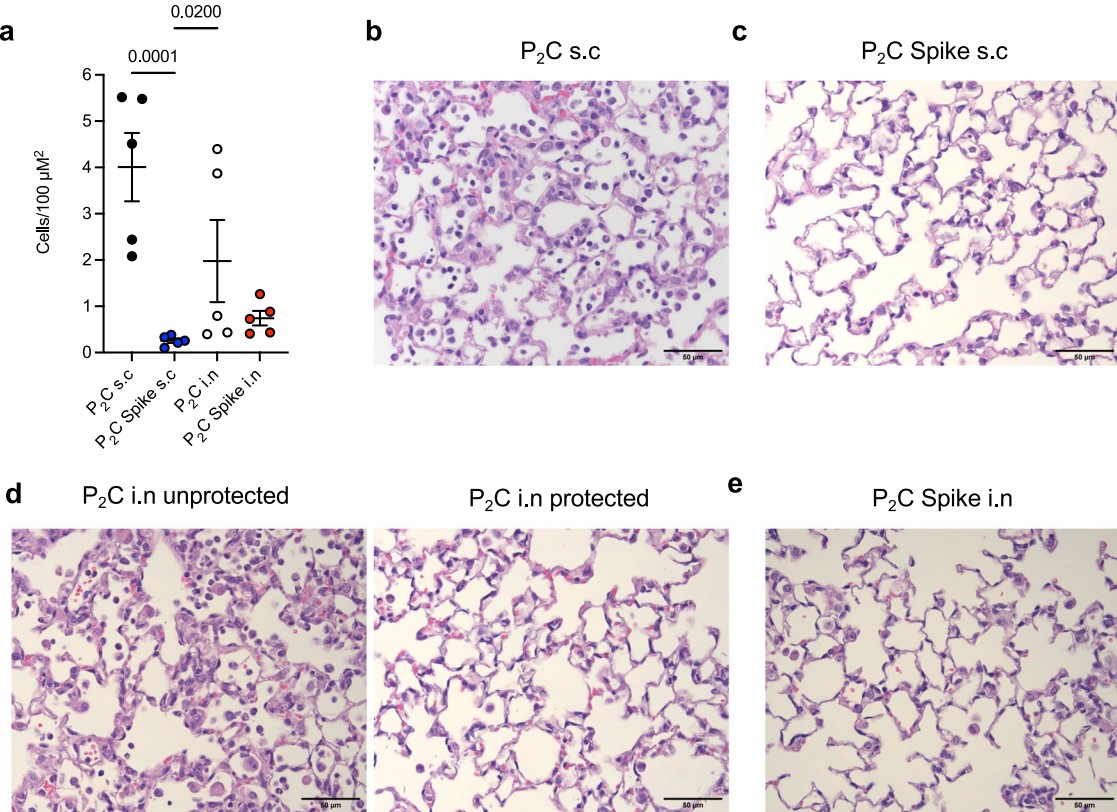

**Fig. 9 | Pam₂Cys Spike vaccination protects mice from damaging leukocyte infiltration of the lung parenchyma during SARS-CoV-2 infection.**
**a** Quantitation of inflammatory cells in the lung parenchyma of vaccinated mice 7 days p.i, Pam₂Cys alone (P₂C; black), or Pam₂Cys Spike, sub-cutaneously (s.c; blue) or intra-nasally (i.n; red) Data are for individual mice ($n = 5$ each), means +/− SEM are shown. Statistics: $p$ values indicated, Kruskal–Wallis test, Dunn's multiple comparisons. Representative lung histological sections from mice 7 days p.i with SARS-CoV-2, vaccinated with (**b**) Pam₂Cys s.c, (**c**) Pam₂Cys Spike s.c, (**d**) Pam₂Cys i.n, a representative for both unprotected and protected phenotypes are shown, and **e** Pam₂Cys Spike i.n. Scale bar = 50 μm. Source data are provided as a Source Data file.

divergent responses, with 60% of mice displaying minimal inflammation which was comparable to those mice that were vaccinated with Pam₂Cys Spike by either route (Fig. 9). Thus, mucosal and systemic vaccination with Pam₂Cys Spike induced potent protective immunity in mice, with evidence of a non-specific beneficial effect of mucosal Pam₂Cys at this early time point after vaccination, a phenotype which has also been observed after i.n TLR2 activating lipopeptide in murine models of rhinovirus and influenza virus infection[33–35].

To evaluate whether sustained sterilizing immunity was maintained post vaccination, including the observed non-specific protective effect of Pam₂Cys i.n alone, mice were vaccinated 3 times ($t = 0$, 2.5 and 5 weeks) before being challenged with SARS-CoV-2 at six weeks post-final vaccination (i.e., at week 11). The maintenance of high titres in the serum of anti-Spike IgG by either route was confirmed one week prior to challenge, with anti-Spike IgA again only detected in mucosally immunised mice (Fig. 10a). Critically, at this longer time point after challenge, complete protection from disease was maintained in mice vaccinated with Pam₂Cys Spike via either route, as measured by the absence of weight loss (Fig. 10b) and clinical symptoms (Fig. 10c). Additionally, sterilizing immunity was maintained in the lungs of Pam₂Cys Spike immunized mice (Fig. 10d), whereas the short-term non-specific beneficial effect of Pam₂Cys i.n at three weeks post vaccination was lost, demonstrating the requirement for pre-existing antigen-specific immunity for protection from infection.

## Discussion

The ability to prevent infection and spread of SARS-CoV-2, or indeed other respiratory viruses of concern such as pathogenic coronaviruses and influenza, is critical to address with vaccine development. Vaccines that only partially protect against infection, coupled with large partially immunized populations, may provide a perfect evolutionary driver for VOCs that evade vaccine immune responses with increased ability to replicate in the respiratory mucosa. It may be difficult for parenterally administered vaccines to provide complete protection from infection owing to the systemic nature of the responses induced[1]. The generation of mucosal immunity, in the URT where viral replication first occurs, requires either local antigen exposure to generate mucosal immune responses, or very high plasma nAb titres that are unlikely to be maintained long term following parenteral vaccination[12]. Analysis of mucosal immunity in recipients of COVID-19 mRNA vaccines revealed that they developed lower levels of SARS-CoV-2 nAbs with no evidence of Spike protein-specific T or B cells in the BALF, compared to the BALF antibody and cellular responses in subjects recovering from COVID-19[36]. Moreover, even convalescent patients do not consistently show clear evidence of protective mucosal immune responses despite having plasma antibodies, highlighting the distinct tissue compartmentalization of immune responses to SARS-CoV-2[37]. Mucosal vaccines, either in isolation or as boosters for parenteral vaccination, may provide a powerful strategy to generate local protective responses in the pulmonary environment, with the potential to reduce the risk of infection and transmission.

Here, we demonstrate that vaccination with a novel protein-based subunit vaccine, Pam₂Cys Spike administered either parenterally or mucosally, led to substantial serum anti-Spike IgG titres and high nAb against SARS-CoV-2 virus, including Beta and Delta variants (Figs. 1 and 8a, b). In mice expressing human ACE2, this resulted in complete

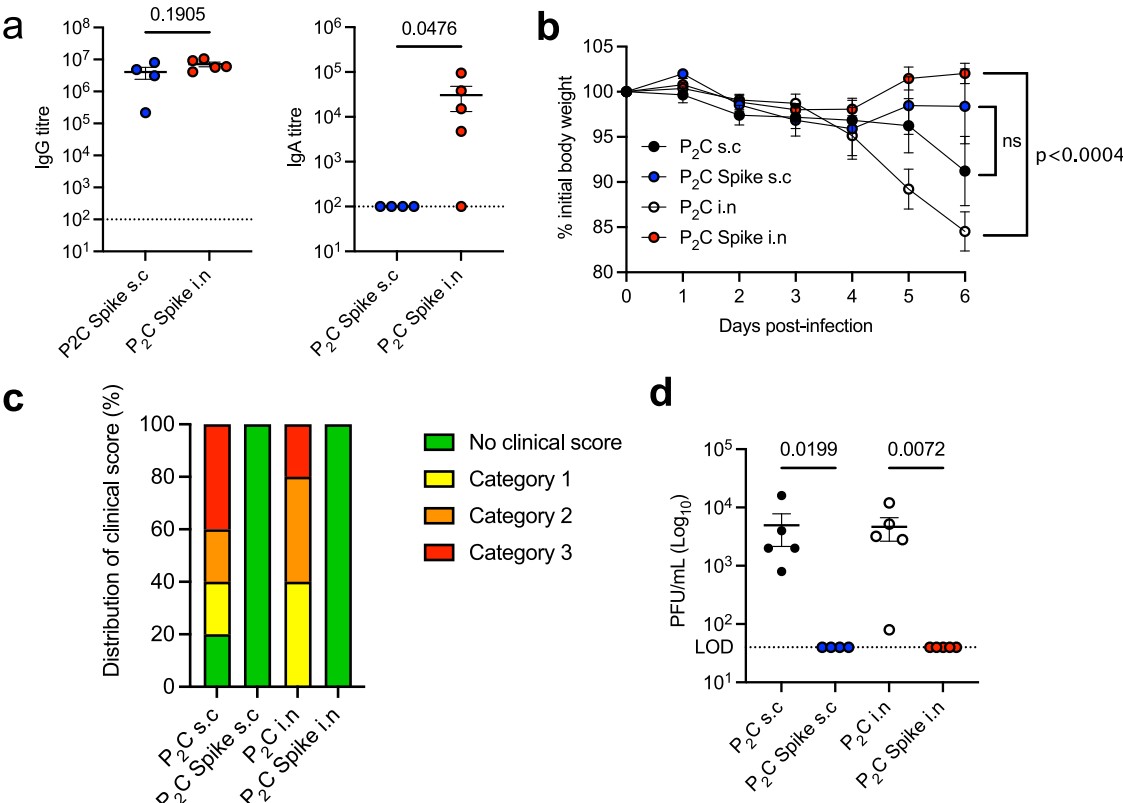

**Fig. 10 | Vaccination with Pam$_2$Cys Spike induces potent sustained protection against SARS-CoV-2 challenge.** K18-hACE2 mice ($n = 5$) were immunized with Pam$_2$Cys alone (P$_2$C; black), or Pam$_2$Cys Spike, sub-cutaneously (s.c; blue) or intra-nasally (i.n; red) three times, 2.5 weeks apart. 6 weeks post final vaccination mice were challenged with SARS-CoV-2 i.n. **a** Anti-spike IgG and IgA titres in serum collected 5 weeks post final vaccination, as determined by ELISA. **b** Mouse weight loss from days 0–6 post infection (p.i) with SARS-CoV-2. **c** Clinical score distribution, increasing category indicates severity, (no clinical score, green; category 1, yellow; category 2, orange; category 3, red) (**d**) viral load (plaque forming units; PFU) at day 6 p.i. Limits of detection (LOD) indicated by the dotted lines. Data are for individual mice ($n = 5$ each), means $+/-$ SEM are shown. Statistics: (**a**) Welch's $t$ test, (**b**) Two-way ANOVA with Tukey's multiple comparisons, (**d**) Kruskal–Wallis test, Dunn's multiple comparisons. $p$-values are indicated. Source data are provided as a Source Data file.

protection against viral infection in the lung tissue and from clinical indicators of disease (Figs. 8, 9, 10). There is strong evidence for a correlation between high serum nAb titres and protection from COVID-19[38], including against VOCs, supporting the benefits of booster vaccinations to maintain high serum nAb titres[39]. Studies in non-human primates demonstrate a correlation between increasing serum nAb titres and protection in the URT, so that lower nAb titres are sufficient to protect the lung alone, compared to higher nAb levels needed to protect both the URT and lungs[40]. Thus, while many of the currently deployed intramuscular vaccines can prevent SARS-CoV-2-induced pneumonia and severe COVID-19, there is greater variability in protection against URT infection and transmission. Importantly, Pam$_2$Cys Spike mucosal vaccination generated high levels of anti-Spike IgA, as well as increased nAb in both the serum and airways (Figs. 1 and 8a, b). Mucosal IgA reduces colonisation of the mucosa by other respiratory pathogens[2], including when induced by i.n vaccination[30,41]. There is scope to explore the addition of defined Spike epitopes to the vaccine that may induce cross-reactive nAb against SARS-CoV-2 variants, or other pathogenic coronaviruses[42]. Mucosal vaccination also increased lung Spike-specific CD4$^+$ T-cell responses (Fig. 3) and lung T$_{RM}$ (Fig. 4), that may contribute to the maintenance of protection. Inclusion of additional viral antigens, such as the nucleocapsid protein, to enhance T-cell immunity through broader epitope coverage[43,44], may sustain long-term protective memory responses in the mucosa, including in individuals with antibody deficiency. This also has the potential to provide cross-protection against other coronaviruses.

TLR2, the pattern recognition receptor activated by Pam$_2$Cys in this vaccine, is widely distributed in the pulmonary environment,

including expression by both respiratory epithelia and antigen presenting immune cells[24,25]. However, the relative contribution of these compartments to the mucosal vaccine response has not been defined. In our work, mice deficient in TLR2 on both the epithelia and hematopoietic compartments had minimal responses to mucosal Pam$_2$Cys Spike vaccination, indicating that vaccine-induced immune activation was dependent on TLR2 stimulation. After TLR2 stimulation of the pulmonary epithelia, IL-6, TNF and IL-1 may be released to recruit neutrophils and activate APCs[22–25]. Interestingly, we demonstrate here that the induction of early neutrophil responses in the lungs and airways was facilitated to the same extent by TLR2 expression on either the epithelia or immune cells (Fig. 5c). TLR2 expression on the epithelia provided an advantage for APC recruitment, including DC, alveolar macrophages and monocyte subsets (Fig. 5d). Nevertheless, the expression of TLR2 in the hematopoietic compartment was required for the optimal generation of lung-localised, antigen-specific adaptive responses after booster vaccination (Figs. 6 and 7), in particular for the strong IL-17$^+$ CD4$^+$ T-cell response, which is typical following mucosal delivery of protein-based vaccines[45–47], particularly those activating TLR2[26,27]. This lung Th17 response was balanced with a Tfh response in the draining lymph nodes (Fig. 7c) and was not associated with evidence of inflammatory damage to the lungs following SARS-CoV-2 infection (Fig. 9).

Of interest, SARS-CoV-2 challenge following parenteral and mucosal immunization revealed that the delivery of the adjuvant Pam$_2$Cys by the i.n, but not the s.c route, resulted in a partial protective effect at early time points (Fig. 8). This suggests that activation of TLR2 in the nasal and lung environment can induce antigen-independent

innate immunity, although this effect was lost by 6 weeks post vaccination (Fig. 10). This expands the observation that a Pam$_2$Cys-based, TLR2 activating lipopeptide could activate the nasal epithelium and immune cells to provide short-term, non-specific protection against rhinovirus and influenza virus infection[33–35]. The method of delivery and volume of the vaccine may influence whether the immune activation occurs in the upper airways alone or the upper and lower respiratory tract. This is important for consideration of clinical translation, as i.n vaccination in humans typically delivers the vaccine primarily to the upper airways[12]. Prophylactic i.n delivery of a TLR2 agonist has recently been demonstrated to protect against URT viral shedding of SARS-CoV-2 in a ferret model[48], and early clinical trials are underway to assess safety and tolerability in humans (ACTRN12621000607875p). This provides further support for Pam$_2$-Cys as a promising mucosal adjuvant for enhancing both local innate defences and induction of powerful adaptive immunity.

Currently, at least eight mucosal vaccines for COVID-19 have entered clinical trials, delivered either exclusively intra-nasally or by aerosol, or as a mucosal booster following parenteral prime[49]. Only one of these is a mucosal protein-based vaccine (utilising a receptor-binding domain recombinant protein without adjuvant, clinical trial RPCEC00000345), with the others being live-attenuated or viral vector-based formulations designed to express the Spike protein[12]. The induction of immunity to the viral vector itself will limit the use of the latter vaccines to boost immunity. As there will be an ongoing requirement for boosting the waning nAb levels to currently used COVID-19 vaccines, subunit protein-based vaccines with safe and effective adjuvants, such as Pam$_2$Cys Spike, will provide a safe alternative for delivery to the nasal mucosa in a needleless fashion. It will also be important to utilize Spike-antigen that will provide Spike-specific responses against the current SARS-CoV-2 variants.

Future studies will require an assessment of the durability of protective responses achievable with this vaccination approach, including in the URT such as the nasal mucosa and trachea, for which SARS-CoV-2 demonstrates a notable tropism[50]. In animal models of influenza, resident memory cells generated in the nasal passages show potential for preventing virus dissemination to the lung[51]. In addition, other animal models, such as hamsters or ferrets, could be used to determine the impact of vaccination on transmission to naïve animals[31,52]. As a large proportion of the world's population has received an intramuscular vaccination for SARS-CoV-2, it will be important to test a "prime-pull" vaccination approach[12,53]. Memory T-cells induced by parenteral vaccination can be recruited to mucosal sites by subsequent mucosal boosting[54], providing a strategy to optimise both systemic IgG and T-cell responses that protect the lower respiratory tract from severe disease, as well as appropriate upper mucosal immunity to prevent infection. Thus, the mucosal vaccination strategy described here has the potential to generate IgA-secreting memory B-cells and tissue resident T-cells in the respiratory tract, to provide protection from viral infection and reduce the risk of ongoing viral transmission with pandemic potential.

## Methods

### Vaccine production
Vaccine adjuvant was Pam$_2$Cys-SK$_4$-triethylene glycolate (Pam$_2$Cys) as previously described[27,55]. SARS-CoV-2 Spike HexaPro expression vector was a gift from Jason McLellan (Addgene plasmid # 154754; http://n2t.net/addgene:154754; RRID:Addgene_154754)[56]. The HexaPro variant contains six beneficial proline substitutions providing ~10-fold higher expression than the parental construct and increased stability of the trimer in prefusion conformation. Hrv3C-His-Strep tags were included for purification purposes. Recombinant Spike protein was expressed in suspension-adapted Expi293F™ cells, harvested, and purified by immobilised metal affinity chromatography (IMAC) using Ni-NTA agarose as previously described[57]. The eluate following IMAC was

dialysed to 2 mM Tris, pH 8.0, 200 mM NaCl, concentrated before being passed through a 0.22 μm filter, and subjected to gel filtration chromatography using HiLoad 16/600 Superdex 200 (Cytiva) in 2 mM Tris, pH 8.0, 200 mM NaCl. Doses per vaccination were as follows: 7.7 μg Pam$_2$Cys-SK$_4$-triethylene glycolate and 6 μg Spike, admixed in sterile cell culture grade phosphate buffered saline, endotoxin tested (PBS; Sigma D8537).

### Mouse strains and immunization procedures
All animal experiments were conducted in full compliance with local and institutional guidelines, with approvals from the Sydney Local Health District Animal Welfare Committee (2020-003, 2020-019). These adhere to the Australian Code for the Care and Use of Animals for Scientific Purposes (2021) as set out by the National Health and Medical Research Council of Australia. Mice had access to standard rodent chow and water ad libitum. All mice were housed under strict husbandry conditions including the use of IVC cages and sterilised bedding, food and water. Mice were on a 12-h light/dark cycle, temperature was maintained at approximately 21 °C and a relative humidity of 45–46%. Mice were provided with tissues for nesting material and sterile cardboard and shelter for environment enrichment. Female C57BL/6 (6–8 weeks-old at start of experiments) or BALB/c (6–17 weeks old) mice were sourced from Animal BioResources (Mossvale, Australia) and housed at the Centenary Institute under specific-pathogen-free conditions. Female Tlr2–/– mice (BALB/c background, 6–17 weeks) were bred at the Centenary Institute[24,25]. Female K18-hACE2 mice[58] were sourced from the Jackson Laboratory and bred at Australian BioResources and the Centenary Institute (8–12 weeks at start of experiments). For s.c injection, mice were anaesthetized with gaseous isoflurane (4%, O$_2$ 1 L/min) and injected at the base of tail with vaccine in a maximum of 200 μl volume. For i.n instillation, mice were either anaesthetized by intraperitoneal injection of ketamine/xylazine (50 mg/6.25 mg/kg) or gaseous isoflurane, then vaccine in 30–50 μL of PBS was applied to the nares such that mice inhaled the solution. For assessment of early innate responses mice were immunized once; for immunogenicity or protection studies mice were immunized three times 2 weeks apart and adaptive immune responses assessed 1 or 8 weeks after the last booster vaccination.

### Generation of TLR2 chimeric mice
WT BALB/c or Tlr2–/– female mice (between 6–17 weeks of age, similar aged WT and Tlr2–/– mice were compared in each experiment) were irradiated at a caesium source (800 RAD). Femurs were isolated aseptically from donor BALB/c or Tlr2–/– female mice into complete culture media, RPMI Glutamax (Gibco) with 10% FBS (Hyclone) and penicillin-streptomycin (100 U/ml; Gibco). Marrow cells were flushed from the femur with a 25 G needle into complete media, 70 μm filtered and erythrocytes lysed with red blood cell lysing buffer Hybri-Max (Sigma). Cells were 70 μm filtered again before washing and suspending in PBS for i.v injection of 1.8 × 10$^6$ cells (in 200 μl) per mouse. Mice were rested for 12 weeks to allow hematopoietic reconstitution. For the first 3 weeks post irradiation, mice received prophylactic antibiotics (Trivetrin; final concentration in drinking water 0.54 mg/ml trimethoprim and 2.7 mg/ml sulfadoxine) and extra dietary supplements (sunflower seeds, DietGel® Boost - Clear H$_2$O) as supportive care.

### Organ collection and processing
Mice were euthanized by CO$_2$ asphyxiation, tissues collected aseptically and maintained at 4 °C. Blood was collected by rupture of the inferior vena cava/portal vein and allowed to clot, before collection of sera after centrifugation (2000 x g, 15 min). BALF was obtained by tracheal intubation, inflation of the airways with 1 ml PBS and collection of the fluid. Cells were isolated by centrifugation, and BALF was immediately frozen to preserve for further analyses. Other tissues were

collected into RPMI 1640 (Gibco) supplemented with 10% FCS (Sigma), 2-ME (0.05 mM), and penicillin–streptomycin (100 U/mL; Gibco). For lung tissue, circulating blood was removed by perfusion with PBS and heparin (20 U/mL; Sigma) injected into the right atrium of the heart. Lung lobes were diced and digested at 37 °C for 45 min with collagenase type 4197 (50 U/mL; Sigma) and DNase I (13 μg/mL; Sigma), then filtered through a 70 μm sieve. Mediastinal lymph nodes and spleens were pressed through a 70 μm filter and the leukocytes pelleted by centrifugation. Erythrocytes were lysed with red blood cell lysing buffer Hybri-Max (Sigma), and leukocytes enumerated with Trypan Blue (0.04%) exclusion by hemocytometer.

## Quantitation of anti-spike antibody isotypes

Spike-specific Ig in serum and BALF were determined by indirect ELISA. High-binding ELISA plates (Corning Falcon) were coated overnight at 4 °C with Spike protein (1 μg/ml) in carbonate/bicarbonate coating buffer (0.05 M, pH 9.6). Plates were washed with PBS (POCD) plus 0.05% (v/v) Tween 20 (Sigma), then blocked with 1% (w/v) BSA (Bovogen) in PBS (1 h at 37 °C). Dilution series of sera (from 1:100) or BALF (from 1:40) were added (1 h at 37 °C) to capture Spike-specific antibody, and then after washing, detected with either horseradish peroxidase-conjugated goat anti-mouse IgG (Invitrogen Novex, cat# A16090, used at 1:2000), goat anti-mouse IgG1 (*Jackson ImmunoResearch* cat#*115-035-205*, used at 1:2000) goat anti-mouse IgG2c (*Jackson ImmunoResearch cat#115-035-208*, used at 1:2000), goat anti-mouse IgA (Invitrogen, cat# 626720, used at 1:2000), or rat anti-mouse IgE (*Abcam, cat# ab99574, clone 23G3, used at 1:2000*). Washed plates were developed with tetramethylbenzidine substrate (Sigma), stopped with 2 M HCl, and then absorbance read at 450 nm with a 570 nm reference wavelength (Tecan Infinite M1000 PRO). The titre was estimated by the titration curve of each sample interpolated with the mean absorbance (+ 3 SD) of the relevant negative samples from unvaccinated or adjuvant only immunized mice (1:100 sera, or 1:40 BALF).

## Cell lines for infection assays

HEK293T (ATCC CRL-3216) and VeroE6 (ATCC CRL-1586) cells were maintained in Dulbecco's Modified Eagle Medium (DMEM; Invitrogen) or Minimal Essential Media (MEM; Invitrogen) supplemented with 10% FBS and 100 μg/mL Penicillin-Streptomycin (Gibco), and sub-cultured according to the supplier's instructions. HEK293T cells stably expressing human ACE2 and/or TMPRSS2 were generated by transducing cells with lentiviral particles and clonal selection as previously described[59]. Cell lines were routinely checked for mycoplasma contamination (MycoAlert™ PLUS Mycoplasma Detection Kit; Lonza) and confirmed as negative.

## Quantitation of nAb titres using a pseudovirus assay

nAbs were quantitated against replication-deficient SARS-CoV-2 ancestral (Wuhan) Spike pseudotyped lentivirus particles as previously described[60]. Briefly, pseudovirus particles were generated by co-transfecting a Spike expression construct pCG1-SARS-2-S Δ18[61,62] and GFP-luciferase vector with lentivirus packaging and helper plasmids into HEK293T cells, using Fugene HD (Promega)[63]. Pseudovirus particles were incubated with serially diluted serum or BALF samples at 37 °C, 5% $CO_2$ for 1 h prior to spinoculation (800 x $g$, 35 °C) of ACE2 over-expressing HEK293T cells. Seventy-two h post-transduction, cells were fixed and stained with Hoechst 33342 (NucBlue™ Live ReadyProbes™ Reagent, Invitrogen) as per the manufacturer's instructions, and imaged using the Opera Phenix high content screening system (Perkin Elmer). The percentage of GFP positive cells was enumerated (Harmony® high-content analysis software, Perkin Elmer) and neutralising endpoint titre determined as the dilution required for ≥50% inhibition of infection ($EC_{50}$), estimated by sigmoidal curve and interpolation in GraphPad Prism software.

## Quantitation of nAb titres against SARS-CoV-2

High-content fluorescence microscopy was used to determine the neutralising capacity of serum and BALF from vaccinated mice to inhibit live SARS-CoV-2 ancestral virus (A.2.2) infection and the resulting cytopathic effect in permissive cells (VeroE6, final MOI = 0.05), analysed with InCarta Software (Cytiva) as previously described[61]. Alternatively, Beta (B1.351), Delta (B.1617.2) or Omicron (BA.5) variants were compared against ancestral virus (A.2.2) in HEK293 cells overexpressing ACE2 and TMPRSS2[59,61]. The neutralisation endpoint titre of diluted serum or BALF samples was set to ≥ 50% neutralisation ($EC_{50}$), estimated by sigmoidal curve and interpolation in GraphPad Prism software. All work utilising SARS-CoV-2 virus was performed in PC3/BSL3 facilities.

## Flow cytometry and ex vivo antigen recall of leukocytes

Lung, BALF, spleen or mediastinal lymph node single cell suspensions were immunolabelled to facilitate phenotyping and data acquired by flow cytometry. Briefly, after washing in PBS, cells were incubated with live/dead fixable blue dead cell stain (Invitrogen) and Fc receptors blocked with anti-mouse CD16/CD32 (2.4G2; BD Biosciences). After washing in FACS wash (PBS with 2% FCS and 2 mM EDTA), surface markers were labelled. For lung and BAL immune profiling, antibodies were anti-mouse B220-BUV737 (RA3-6B2; BD), SiglecF-BV421 (E50-2440; BD), MHCII-V500 (M5/114.15.2; BD), Ly6G-BV650 (1A8; Biolegend), CD11b-BV785 (M1/70; Biolegend), CD45.2-FITC (104; BD), CD11c-PE (N418; Biolegend), CD3-PECF594 (145-2C11; BD), CD64-PECy7 (X54-5/7.1; Biolegend), TLR2-AF647 (6C2; BD), CD4-AF700 (RM4-5; Biolegend) or CD4-AF700 (GK1.5; eBioscience), and Ly6C-APCCy7 (HK1.4; Biolegend). For detection of $T_{RM}$-like T-cells in the lungs, antibodies were B220-BUV737 (RA3-6B2; BD), CD62L-BV421 (MEL-14; BD), CD103-FITC (2E7; Biolegend), CD44-PE (IM7; Biolegend), CD3-PECF594 (145-2C11; BD), CD69-PECy7 (H1.2F3; BD), CD45.2-APC (104; BD), CD4-AF700 (RM4-5; Biolegend) and CD8-APCCy7 (53-6.7; BD). For mediastinal lymph nodes, cells were first incubated with Fc receptor blocking anti-mouse CD16/CD32 (2.4G2; BD Biosciences) and anti-CXCR5-biotin (2G8; BD) at 37 °C for 40 min, washed in PBS, then incubated with live/dead fixable blue dead cell stain (Invitrogen) and streptavidin-PE (BD) in PBS for 20 min at 4 °C. Cells were then surface immunostained with anti-mouse PD1-BV785 (29F1A12; Biolegend), CD3-PECF594 (145-2C11; BD), CD44-PECy7 (IM7; BD), CD4-AF700 (RM4-5; Biolegend) or CD4-AF700 (GK1.5; eBioscience), and CD8-APCCy7 (53-6.7; BD). Transcription factor staining with anti-mouse Bcl6-AF647 (K112-91; BD) was performed using eBioscience FoxP3/Transcription Factor staining buffer set (Invitrogen), according to the manufacturer's instructions.

Enumeration of Spike-specific T-cell responses was performed by ex vivo antigen recall, intra-cytoplasmic staining and flow cytometry. Lung or spleen leukocytes were stimulated for 4 h (37 °C, 5% $CO_2$) with Spike protein (5 μg/ml) and/or $CD8^+$ epitope peptide Spike$_{338-346}$ (Genscript), or for controls media alone, or anti-mouse CD3 (1452C11; 5 μg/mL) and anti-mouse CD28 (37.51; 5 μg/mL; BD Pharmingen). Brefeldin A (10 μg/mL; Sigma) was added and further incubated such that cytokine accumulated intracellularly (12 h, 37 °C, 5% $CO_2$). Viability stain and Fc blocking was performed, then surface markers were immunolabelled with anti-mouse CD8-APCCy7 (53-6.7; BD Pharmingen), CD4-AF700 (RM4-5; Biolegend) or CD4-AF700 (GK1.5; eBioscience) and CD44-PECy7 (IM7; BD) and then cells were washed thoroughly. Cells were fixed with BD Cytofix/perm followed by thorough washing with BD Perm/Wash. Intracellular cytokines were labelled with anti-mouse IFNγ-FITC (XMG1.2; BD Pharmingen), IL-17A-PB (TC11-18H10.1; Biolegend), IL-4-PE (11B11; BD), TNF-PerCPCy5.5 (MP6-XT22; BD), and IL-2-APC (JES6-5H4; Biolegend) prepared in BD Perm/Wash buffer and then washed. Compensation controls were prepared in the same manner as experimental samples, viability stain was prepared from murine leukocytes, and UltraComp eBeads Compensation

Beads (Invitrogen) were stained with the same antibody used in experimental panels. Samples were acquired using an LSRII 5 L flow analyzer (BD Biosciences) using BD FACSDiva software.

Computational analysis of lung leukocytes was performed using the Spectre R package[32], with instructions and source code provided at https://github.com/ImmuneDynamics/spectre. Compensated sample data were initially prepared in FlowJo (BD) with quality control gating to exclude debris and dead cells, then the CD45+ population was exported as raw value CSV files. Arcsinh transformation was performed on the data in R using a co-factor of 1,000 to redistribute the data on a linear scale and compress low end values near zero. The dataset was then merged into a single data.table, with keywords denoting the sample, group and cell numbers. The FlowSOM algorithm[64] was then run on the merged dataset to cluster the data into 30 clusters, where every cell is assigned to a specific cluster and metacluster. Subsequently, the data were downsampled and analysed by the dimensionality reduction algorithm FIt-SNE for cellular visualisation[65]. Alternatively, flow cytometry data were analysed with FlowJo (v10; BD), see Figs. S2, 5, 11 and 12 for gating strategies.

Cytokines in the BALF of vaccinated or control mice were determined by Legendplex (mouse Th cytokine panel 13-plex; Biolegend). Data were acquired on an LSR Fortessa X20 (BD) and analysed using Biolegend software.

### SARS-CoV-2 murine challenge model and organ collection

Mice were housed in a PC3/BSL3 facility in an IsoCage N biocontainment system (Tecniplast, Italy). Three or six weeks after final i.n vaccination, female K18-hACE2 mice were anaesthetised with gaseous isoflurane and infected i.n with $10^3$ plaque forming units (PFU) of SARS-CoV-2 (VIC01/2020) in 30 µl final volume as previously described[66]. Clinical indicators of disease, including weight, were monitored for the following 7 days. Clinical signs were classified into four categories based on the following criteria: No clinical score - mice did not exhibit any clinical symptoms; Category 1 – weight loss, early lethargy and hunching; Category 2 – substantial weight loss, lethargy, hunching, ruffling and early laboured breathing; Category 3 – all signs in Category 2 are exacerbated, accompanied by immobility. Mice were euthanised with intraperitoneal overdose of pentobarbitone (Virbac, Australia). The multi-lobe lungs were tied off using surgical thread and BALF was collected from the single lung lobe via lavage with 1 mL HANKS solution (Sigma-Aldrich, USA) with a blunted 19-gauge needle inserted into an incision above the largest cartilage ring of the trachea. BALF was centrifuged (300 x g, 4 °C, 7 min), and supernatants were collected and snap frozen. The remaining cell pellet was resuspended in 200 µL Red Blood Cell Lysis Buffer (ThermoFisher, USA) for 5 min, followed by addition of 700 µL HANKS solution and then centrifuged again. Following centrifugation, the supernatant was discarded and the cell pellet was resuspended in 160 µL HANKS solution and enumerated using a haemocytometer (Sigma-Aldrich, USA). Differential BALF counts were performed by depositing 70 µL of BALF leukocytes into a cytospin funnel and then centrifuging onto a glass slide (300 x g, 7 min). Slides were left to airdry overnight, after which they were stained using Quick Dip Stain Kit (Modified Giemsa Stain) protocol as per manufacturer's instructions (POCD Scientific, Australia). Differential counts were determined based on cellular morphology and staining using light microscopy with a Primovert Zeiss Microscope under a 20X objective (Zeiss, Germany). Multi-lobe lungs were collected and snap frozen on dry ice. Lung homogenates were prepared fresh, with the third lobe of the multi-lobe lungs placed into a gentle MACS C-tube (Miltenyi Biotec, Australia) containing 2 mL HANKS solution. Lung tissue was homogenised using a gentleMACS tissue homogeniser, followed by centrifugation (300 x g, 7 min) to pellet cells, and

collection of supernatants for plaque assays. The single lobe lungs were perfused with 0.9% NaCl solution via the heart, followed by inflation with 0.5 mL 10% neutral buffered formalin (Sigma-Aldrich, USA) through the tracheal incision, and placed into a tube containing 10% neutral buffered formalin. Lung lobes were fixed in formalin for at least 2 weeks after which, single lung lobes were transported to a PC2 facility for paraffin-embedding prior to being cut into 3 µm thick sections using a Leica microtome (Leica, Germany). Histology sections were stained using Quick Dip Stain Kit (Modified Giemsa Stain) protocol as per manufacturer's instructions (POCD Scientific, Australia). Inflammatory cells in single lobe lungs were counted using a Zeiss Axio Imager.Z2 microscope with a 40X objective (Zeiss, Germany).

### Plaque assays

Plaque forming units (PFU) of SARS-CoV-2 viral titres were quantitated in VeroE6 cells[66]. Cells were deposited into a 24-well cell culture plate at $1.5 \times 10^5$ cells/well in a 0.5 mL volume and allowed to adhere overnight. The next day, virus-containing samples from lung homogenates were serially diluted in MEM cell culture medium and cell culture supernatants removed from the VeroE6 cells and 250 µL of virus-containing samples was added to the surface of the cell monolayers. Plates were placed in the cell culture incubator (37 °C/5% $CO_2$) and rocked gently every 15 min to facilitate viral adhesion. Following 1 h incubation, 250 µL of 0.6% agar/MEM solution was overlaid gently onto the cell culture samples and placed into the incubator. At 3 days post-infection, an equal volume of 8% paraformaldehyde solution (4% final solution) was added to all wells and incubated at room temperature for 30 min. Following fixation, the supernatant was discarded and the cell monolayer washed twice with PBS, followed by incubation with 0.025% crystal violet solution (Sigma-Aldrich, USA) for 5 min at room temperature and then two additional washes with PBS prior to revealing viral plaques.

### Statistics

The significance of differences between two experimental groups was evaluated by two-tailed Welch's $t$ test or Mann–Whitney test. Multiple experimental groups were compared by Kruskal–Wallis test or two-way analysis of variance (ANOVA), with pairwise comparisons achieved using Dunn's or Tukey's post-hoc multiple comparisons test. Differences were considered statistically significant when $p \le 0.05$, exact values are indicated.

### Reporting summary

Further information on research design is available in the Nature Research Reporting Summary linked to this article.

## Data availability

Source data are provided with this paper. Full datasets generated during and/or analysed in the current study are available from the corresponding author on reasonable request.

## Code availability

Computational analysis was performed using the Spectre R package[32], with instructions and source code provided at https://github.com/ImmuneDynamics/spectre.

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

## Acknowledgements

This work was supported by grants from the National Health and Medical Research Council APP1153493 (to W.J.B.), APP2011467 (to W.J.B. and M.D.J.), APP1175134 (to P.M.H.) and APP1174941 (to R.J.P.), the Medical Research Future Fund (to M.S. and J.A.T.), the Rainbow Foundation, NSW RNA Biology hub and UTS (to P.M.H.), the University of Sydney Drug Discovery Initiative and the University of Sydney Institute for Infectious Diseases (to A.S.A. and M.S.) and the Kenyon Foundation (to M.D.J.). We acknowledge the assistance of Dr David McDonald in production of adjuvant stocks, Dr Charles Bailey for supply of lentivirus constructs, the support of the Sydney Cytometry Core Research Facility, and the Centenary Institute animal facility. Figs. 1a, 5a and 6a were partly created with BioRender.com.

## Author contributions

A.S.A., J.W.C.M., R.J.P. and W.J.B. initiated and designed the research studies. A.S.A., J.W.C.M., M.D.J., S.S., C.L.A., A.A., R.S., S.M., D.H.N., C.C., S.T., conducted experiments and acquired data. J.W.C.M., R.S., J.P.M., C.C., S.N.B., S.T., M.S., J.A.T., P.M.H. provided reagents and designed or established critical assays utilised. A.S.A., S.S., S.N.B., P.M.H., R.J.P., and W.J.B. interpreted and analyzed data. A.S.A. wrote the manuscript with contributions from all authors.

## Competing interests

The authors declare no competing interests.

## Additional information

Anneliese S. Ashhurst[1,2,3,4,5], Matt D. Johansen[6], Joshua W. C. Maxwell[3,7], Skye Stockdale[1,2], Caroline L. Ashley[1,2], Anupriya Aggarwal[8], Rezwan Siddiquee[9], Stefan Miemczyk[6], Duc H. Nguyen[6], Joel P. Mackay [9], Claudio Counoupas[1,2,4,5], Scott N. Byrne [1,2,10], Stuart Turville [8], Megan Steain[1,2,4], James A. Triccas [1,2,4], Philip M. Hansbro [6], Richard J. Payne [3,7] ✉ & Warwick J. Britton [5,11] ✉

[1]School of Medical Sciences, Faculty of Medicine and Health, The University of Sydney, Sydney, NSW 2006, Australia. [2]Charles Perkins Centre, The University of Sydney, Sydney, NSW 2006, Australia. [3]School of Chemistry, The University of Sydney, Sydney, NSW 2006, Australia. [4]The University of Sydney Institute for Infectious Diseases, Sydney, NSW 2006, Australia. [5]Tuberculosis Research Program Centenary Institute, The University of Sydney, Camperdown, NSW 2006, Australia. [6]Centre for Inflammation, Centenary Institute and Faculty of Science, School of Life Sciences, University of Technology, Sydney, NSW 2007, Australia. [7]Australian Research Council Centre of Excellence for Innovations in Peptide and Protein Science, The University of Sydney, Sydney, NSW 2006, Australia. [8]Kirby Institute, Sydney, NSW 2052, Australia. [9]School of Life and Environmental Sciences, The University of Sydney, Sydney, NSW 2006, Australia. [10]Westmead Institute for Medical Research, Centre for Immunology and Allergy Research, Westmead, NSW 2145, Australia. [11]Department of Clinical Immunology, Royal Prince Alfred Hospital, Camperdown, NSW 2050, Australia. ✉e-mail: richard.payne@sydney.edu; w.britton@centenary.org.au

