## [Peer Review File · Nature Communications]

Mucosal TLR2-activating protein-based vaccination induces potent pulmonary immunity and protection against SARS-CoV-2 in miceReviewers' Comments:

Reviewer #1:

Remarks to the Author:

The report from Ashhurst et al., shows the results of studies using a recombinant SARS-CoV-2 spike protein antigen with a TLR agonist as a mucosal adjuvant. The results suggest that an intranasal route of immunization with the TLR2 adjuvant is not just effective in promoting mucosal IgA responses, but also that these responses are effective in neutralizing the virus using a variety of assays of infection. The additional studies on TLR2 knockouts and bone marrow chimeras also indicate that TLR2 expression on both epithelial and hemopoietic cells supports the response but hemopoietic expression is more important.

In general, there is no surprise that intranasal administration of an antigen is capable of inducing IgA responses (regardless of adjuvant) and that they are neutralizing. The addition of TLR2 agonist adjuvant is a helpful addition, and the rationale for this approach is sound, and the data support the choices made here. The studies go to a lot of detail on profiling the various cells in the lung for TLR2 expression and so on, and it is not completely clear that the elaborate clustering of the flow cytometry profiles adds much to the analysis here. Given the mucosal site, adjuvant and delivery, it is disappointing that some key information was not assessed, such as specific IgG isotypes (e.g., those more likely to be associated with Th1 vs Th2 responses). Also, IgE titers and potential allergic Th2 type cytokines were not assessed.

A lot of key details were not included in this study, such as the question of whether TLR2 agonist is any more effective than any other adjuvant, such as other TLR2 agonists (e.g., targeting TLR2/6 or TLR2/1 dimers), TLR4 or TLR5 agonists which could also affect epithelial cells, or another B cell-inducing adjuvant compound. A simple mixing of the antigen and adjuvant appears adequate, though there are no additional studies to assess the impact of the specific formulation, excipients, and delivery. Moreover, while this combination is effective in inducing a good IgA response, there is much more exploration in store to assess the persistence of the response, any effect of the adjuvant on affinity maturation, and so on, though clearly these are beyond the scope of this study. But in general the impact of this study is limited by the lack of comparison to alternative approaches.

Interestingly, some of the data suggest a minor effect of intranasal TLR2 agonist even without the virus antigen, and this is perhaps even more interesting in light of the effect on infiltrating cells in the K18-hACE2 mouse infections, which may indicate some separate impact of innate immune activation on innate resistance to infection, and this is of potential interest on its own. It might be worth adding a reference to this effect in the abstract.

This paper will be of interest to those interested in seeing strategies to address ongoing and new infectious disease challenges (especially COVID) and with a few revisions it would be quite suitable for publication. The data presentation is adequate, though some of the figures are a bit busy and include data that adds relatively little to the understanding of the results. Minor revisions might include a more extended discussion of the effect of the TLR2 agonist alone, and a mention in the abstract. Also some mention of the IgG isotype and possible Th1/Th2 cytokine profiles should be included even if the data isn't available.

Reviewer #2:

Remarks to the Author:

In the present study, Ashhurst et al report on a mucosal TLR2-activating protein-based vaccination against SARS-CoV-2. The benefits of mucosal vaccinations in regard to the prevention of transmission and thereby reducing the number of infections are well presented and are of course a very relevant topic at present times. Here, the authors coupled a SARS-CoV-2 spike protein to the TLR-2 agonist

Pam2Cys, a strategy which they applied successfully in a former study on a tuberculosis vaccine. They compared the immunogenicity and efficacy of this vaccine applied either via the s.c or the i.n. route and could confirm that only the i.n. route results in the production of IgA in the Bal and substantial numbers of antigen-specific T-cells in the lung. This is in line with other reports and supports the importance of the route of administration. Since substantial titers of neutralizing antibodies in sera were induced after vaccination at both locations, no differences in the protection against a SARS-CoV-2 infection could be observed in mice. In both cases, sterile immunity is postulated and there are only minor differences in the numbers of infiltrating cells in the lungs after the challenge (Fig.7). Most interestingly, there is still impaired replication and partial protection seen in animals which have received the adjuvant i.n, but not s.c. Unfortunately, this effect is not further evaluated and makes it difficult to conclude on the real contribution of the local antigen-specific immunity induced by spike-Pam2cys. It would be interesting to see, what are the main effectors behind this antigen-unrelated protection? Are there increased levels of cytokines? Is there still an on-going inflammatory response three weeks after the treatment?

In the second part of the study, the authors wanted to analyze the contribution of TLR-2 signaling on hematopoietic and somatic cells for innate and adaptive responses by using BM-chimeras. They vaccinated the different BM-chimeras and analyzed the CD45+ cellular composition in lung and BAL 24h after one immunization or three weeks after the last of three immunizations. Briefly, they report that the early neutrophil infiltration depends on TLR-2 signaling on epithelial cells and the efficient recruitment of CD4 T-cells depends on TLR-2 signaling on CD45+ cells.

Overall, a protein-based vaccine suitable for mucosal vaccinations would be highly desirable and in advantage over vector-based approaches because of the pre-existing immunity and the ability to be used as repetitive booster vaccine. The present study offers some interesting insights in this topic, but also lack some substantial controls to really understand the protective mechanisms.

Major points:

- 1) A non-adjuvanted spike vaccine would be beneficial to judge on the adjuvant effect by Pam2Cys. From the data, it is very obvious that Pam2Cys plays a substantial role in inducing inflammation at the mucosal surface and even provide non-antigen-specific protection, but if this also beneficial for providing long-lasting memory responses is not clear. It would be therefore also necessary to study protection at later time points after the immunization where no unspecific effects of the adjuvant will be present anymore. Overall, the immunization schedule with 3 applications within 6 weeks and an observation period of 3 weeks post boost are rather short and
- 2) The work on the BM-chimera is really time-intensive and laborious, but the results presented here offers only limited insights in the mechanisms and the real contributions of different TLR-2 expressing cell populations. First, there are only this two selected time points (early and late) and second a non-vaccinated or even better a non-adjuvanted control is missing. A non-vaccinated control would be needed to confirm no general differences in the homeostasis of the different CD45+ populations by the missing TLR-2 signal on either the CD45+ or the epithelial compartment. Furthermore, this results are highly descriptive without direct information on the contribution to later observed immune responses. Concretely, is the early influx of neutrophils driven by TLR-2 signaling important for the IgG or IgA or CD4 T-cell response later on? To really generate mechanistic links would be very hard, but at least a kind of kinetic following the early events of cell migration and cytokine expression patterns would be nice to see.
- 3) As mentioned in the summary, it would be highly informative to study the non-specific protection of Pam2Cys alone and also to somehow establish correlates of protection for the vaccine. Since the neutralizing antibody response is so high in both vaccine groups that sterile immunity can be reached, it would be nice to see if the IgA response would be beneficial for neutralization at the mucosal surface, e.g. by neutralization assay with BAL samples against VOCs?
- 4) It is interesting to see that the T-cell response is dominated by Th17 CD4 T-cells and almost no TH1 CD4 or CD8 T-cells are induced. I think, it was expected from the previous TB study, but it would be interesting to see how these responses could contribute to the protection against SARS-CoV-2. This could be at least discusses later on.

Minor points:

5) Although I understand the value of unbiased TSNE-analyses of flow data, here it does not seem to be the best way to present the efficacy of bone marrow reconstitution (Fig. 3+4). From the plots in 3b and 4b, it is not obvious to which content the reconstitution was established and which cells mainly express TLR-2. For that, one has to look in the complicated cluster figures in the supplement. It would be nice to show the TLR-2 expression on selected populations (similar to sup.4 or 6), and estimate the percentage of TLR2+ and TLR2- cells, e.g. CD4+ TLR2+/- or DC TLR2+/- . Only from that, one could judge on the efficient depletion of CD45+ cells by irradiation and exclude residual TLR2+ cells in WT-KO mice.

6) The differences between the cell migration pattern in lung and BAL described for the BM-chimera are very interesting, but there is not much written or discussed about the potential reasons (e.g. different kinetics)

Reviewer #3:

Remarks to the Author:

This is an interesting and informative set of experiments, presented in a scholarly fashion. For the most part, it explores the cell types that respond to the TLR2 agonist, introduced intranasally. Both early innate and later adaptive and cellular recruitment to the respiratory tract are evaluated. The experimental results themselves are comprehensive, well designed and analyzed in a sophisticated manner.

There are several limitations in the analyses, discussed below.

However, a central criticism of this manuscript is that it is a composite of fundamental issues of TLR2 agonist Pam2Cys mediated activation and recruitment of cells to the respiratory tract, what cells are involved in signalling and what events follow ligand recognition, in both the antigen specific and antigen non-specific compartments. The studies here are well designed and well characterized.

But the primary premise of the paper is that this vaccine system will ultimately be useful for protection from SARS-CoV2. It seems that this is an add-on feature of the study to make it more timely. IL17 is the major lung cytokine produced and this cytokine as well as many of the cells that are recruited are well known to primarily protect against bacterial and fungal infections. This cytokine can also lead to lung damage and has a significant risk for proinflammatory responses in humans. Thus, the likelihood of this SARS vaccine strategy seem low.

The authors observed protection from SARS in an B6 animal model, but these data are not compelling. First, the authors use an older transgenic model for ACE-2 expression that does not recapitulate normal expression of this SARS receptor in humans. Second, because this transgenic model is in B6 mice, and the TLR2 bone marrow chimera is developed and explored in BALB/c mice, it is difficult to extrapolate the first half of the paper with the second half. These two strains are dramatically different in T cell phenotype. So the events discovered in one strain cannot be expected to be replicated in another.

Also, and most fundamentally, the events following TLR2 engagement and robust IL17 responses are infiltration of cells that may be useful for bacterial infections but unlikely to be optimal for a viral infection.

Finally, a series of 3 sequential IN prime/boost vaccine regimens is hard to translate to a path in human SARS vaccine design

Other issues:

The T cell phenotype is not fully explored and there is inadequate treatment of cells producing multiple cytokines.

The sources of IL-17 are only discussed in the context of T cells -the authors should determine sources of IL-17 by methods such as direct ex vivo staining.

Nature Communications NCOMMS-21-50205: Resubmission

Ashhurst AS, et al; Mucosal TLR2-activating protein-based vaccination induces potent pulmonary immunity and protection against SARS-CoV-2 in mice.

Response to Reviewers' comments

Reviewer #1:

The report from Ashhurst et al., shows the results of studies using a recombinant SARS-CoV-2 spike protein antigen with a TLR agonist as a mucosal adjuvant. The results suggest that an intranasal route of immunization with the TLR2 adjuvant is not just effective in promoting mucosal IgA responses, but also that these responses are effective in neutralizing the virus using a variety of assays of infection. The additional studies on TLR2 knockouts and bone marrow chimeras also indicate that TLR2 expression on both epithelial and hemopoietic cells supports the response but hemopoietic expression is more important.

In general, there is no surprise that intranasal administration of an antigen is capable of inducing IgA responses (regardless of adjuvant) and that they are neutralizing. The addition of TLR2 agonist adjuvant is a helpful addition, and the rationale for this approach is sound, and the data support the choices made here. The studies go to a lot of detail on profiling the various cells in the lung for TLR2 expression and so on, and it is not completely clear that the elaborate clustering of the flow cytometry profiles adds much to the analysis here. Given the mucosal site, adjuvant and delivery, it is disappointing that some key information was not assessed, such as specific IgG isotypes (e.g., those more likely to be associated with Th1 vs Th2 responses). Also, IgE titers and potential allergic Th2 type cytokines were not assessed.

We thank the reviewer for these helpful suggestions. We have included substantial additional data in this revised manuscript to further characterize the vaccine-induced immune responses:

1. Titration of anti-Spike IgG, IgG1, IgG2c, and IgA at both one and eight weeks post final vaccination, in serum and BALF. This has been added in revised Fig 1b, the new Fig 2, and the data described in results text lines 136-194. We demonstrate that vaccination by either route induces substantial anti-Spike IgG, IgG1 and IgG2c titres in both serum and BALF, but mucosal vaccination induces significantly greater titres in the airways and exclusively anti-Spike IgA.
2. We also assessed Spike-specific antibody and T-cell responses after s.c or i.n vaccination with antigen alone (i.e. Spike protein) versus adjuvanted antigen (Pam₂Cys+Spike). We demonstrate the requirement for the adjuvant, given mucosally, to induce IgA and maximise other antibody and T-cell responses. These results are included in the new Fig 2 and new Fig 3e,f.
3. While ELISA detected total IgE in serum, there was no anti-Spike IgE detectable at any time point post vaccination in either serum or BALF. This result has been included in the text at line 153.
4. To determine whether potential allergic Th2 type cytokines were induced, we performed a T-helper 13-plex Legendplex (Biolegend) on BALF samples at one week post 3-vaccinations, to assess release of inflammatory or Th-associated cytokines at the airways. These data are included in new Fig. S3, and are described in the revised manuscript at lines 217-220. We demonstrate minimal detection of cytokines in the BALF at this timepoint, with a small but insignificant release of IL-17A in the BALF of mucosally immunized mice.
5. We also performed similar Legendplex assays in BALF from *Tlr2*^{-/-} and WT chimeric mice at 24 hours post one mucosal vaccination with Pam₂Cys+Spike, to gain mechanistic understanding of the role of TLR2 responses in early cytokine responses to vaccination at the airway. We observed release of cytokines TNF and IL-6 only in groups of mice that had received wild type hematopoietic cells, consistent with TLR2-mediated activation of alveolar macrophages. These data are included in new Fig. S9, and described in the text of the revised manuscript at lines 341-343.

A lot of key details were not included in this study, such as the question of whether TLR2 agonist is any more effective than any other adjuvant, such as other TLR2 agonists (e.g., targeting TLR2/6 or TLR2/1 dimers), TLR4 or TLR5 agonists which could also affect epithelial cells, or another B cell-inducing adjuvant compound. A simple mixing of the antigen and adjuvant appears adequate, though there are no additional studies to assess the impact of the specific formulation, excipients, and delivery.

We agree with the reviewer that a study performing direct comparisons of TLR2 agonists as adjuvants to other adjuvants, such as TLR4 or TLR5 agonists, and a direct comparison of our vaccine technology to alternative approaches such as viral vector or mRNA vaccines would be valuable for the field. However, this was not the intention of this study and such experiments are out of the scope of this already comprehensive manuscript. The purpose of this study is to explore the mechanism of action of Pam₂Cys as a mucosal adjuvant and to demonstrate the powerful efficacy of this novel vaccination strategy to protect from SARS-CoV-2.

It is also beyond the scope of this first report to do formulation/excipient studies, which would be the focus of future work toward the clinic. That being said, we note that based on our results in this manuscript, PBS or saline provides a suitable buffer system for admixed Pam₂Cys+protein vaccines, and there is no need for additional excipients when this is to be given as a liquid mucosal vaccine. Indeed, we consider that the simplicity of this admix formulation strategy for nasal delivery is actually a benefit of this approach.

As detailed in the Introduction and Discussion sections of the manuscript, our groups have previously compared mucosal adjuvants for other respiratory infections, notably a comparison of TLR2 agonists targeting TLR2/6 vs TLR2/1 dimers, covalently linked to protein antigen (Hanna and Ashhurst et al, *PNAS* 2021), and found similar outcomes in terms of immunogenicity and protection in a model of virulent *Mycobacterium tuberculosis* aerosol infection. Pam₂Cys is easier and cheaper to produce synthetically compared to Pam₃Cys. In addition, as proof of principle, we previously reported a dry powder inhalable protein-TLR2 agonist conjugate vaccine using mannitol as an excipient, which facilitates long term storage at room temperature, and without the need for a cold chain (Tyne et al, *Vaccine* 2013). This has already been described in the Introduction/Discussion and highlights the practical advantages of this approach for either nasal or inhalable delivery.

Moreover, while this combination is effective in inducing a good IgA response, there is much more exploration in store to assess the persistence of the response, any effect of the adjuvant on affinity maturation, and so on, though clearly these are beyond the scope of this study. But in general the impact of this study is limited by the lack of comparison to alternative approaches.

We agree with the reviewer that it is beneficial to examine persistence of vaccine-induced responses, and have included substantial additional data in this revised manuscript, including demonstrating the essential requirement for adjuvant to develop optimum adaptive immune responses to Spike:

1. Titration of anti-Spike IgG, IgG1, IgG2c, and IgA at eight weeks post final vaccination, in serum and BALF. This has been added in the new Fig 2. We demonstrate that vaccination by either route induces substantial anti-Spike IgG, IgG1 and IgG2c titres in both serum and BALF, but mucosal vaccination induces significantly greater titres in the airways and exclusively anti-Spike IgA.
2. Assessment of T-cell responses in the lungs at eight weeks post final vaccination, including Spike-specific CD4⁺ and CD8⁺ T-cell cytokine responses (revised Fig. 3 e,f). We also examined induction of tissue resident memory T-cells in the lungs post vaccination and identified that only mucosal vaccination with Pam₂Cys+Spike induced T_{RM}, particularly CD4⁺ T_{RM}, in the lungs (new Fig 4, described in text of the revised manuscript on lines 224-235).
3. We also assessed Spike-specific antibody and T-cell responses after s.c or i.n vaccination with antigen alone (i.e Spike protein) versus adjuvanted antigen (Pam₂Cys+Spike). We demonstrate the requirement for adjuvanted antigen, given mucosally, to induce IgA and maximise other antibody and lung T-cell responses. This is included in the new Fig 2, revised Fig 3e,f, and the new Fig 4 in the revised manuscript.

Interestingly, some of the data suggest a minor effect of intranasal TLR2 agonist even without the virus antigen, and this is perhaps even more interesting in light of the effect on infiltrating cells in the K18-hACE2 mouse infections, which may indicate some separate impact of innate immune activation on innate resistance to infection, and this is of potential interest on its own. It might be worth adding a reference to this effect in the abstract.

We have now added reference to the early protective effect of i.n TLR2 agonist alone into the abstract, at lines 44-45 as recommended by the reviewer. In addition, we have added additional data to the revised manuscript,

examining protection from SARS-CoV-2 in K18-hACE2 mice challenged at 6 weeks after final vaccination. At this later time point, the innate resistance to infection provided by i.n Pam₂Cys alone was lost, whereas sterilising immunity after Pam₂Cys+Spike vaccination was maintained (data provided in new Figure 10, and discussed on lines 516-528 of the revised manuscript). This highlights that while innate immune activation at the respiratory mucosa may provide some short-term benefit to resist viral infection, antigen-specific immunity is required to maintain protection longer term.

This paper will be of interest to those interested in seeing strategies to address ongoing and new infectious disease challenges (especially COVID) and with a few revisions it would be quite suitable for publication. The data presentation is adequate, though some of the figures are a bit busy and include data that adds relatively little to the understanding of the results. Minor revisions might include a more extended Discussion of the effect of the TLR2 agonist alone, and a mention in the abstract. Also some mention of the IgG isotype and possible Th1/Th2 cytokine profiles should be included even if the data isn't available.

We thank the reviewer for these comments. As detailed above, we have now added additional data to assess i.n TLR2 agonist alone in the protection model at later time points (Fig. 10), results on the IgG isotypes in sera and BALF at one and eight-weeks post vaccination (Fig. 1, 2) and data to profile Th1/Th2 cytokines in BALF 24 hours or 1 week after vaccination (Fig. S3 and S9).

Reviewer #2:

In the present study, Ashhurst et al report on a mucosal TLR2-activating protein-based vaccination against SARS-CoV-2. The benefits of mucosal vaccinations in regard to the prevention of transmission and thereby reducing the number of infections are well presented and are of course a very relevant topic at present times. Here, the authors coupled a SARS-CoV-2 spike protein to the TLR-2 agonist Pam₂Cys, a strategy which they applied successfully in a former study on a tuberculosis vaccine. They compared the immunogenicity and efficacy of this vaccine applied either via the s.c or the i.n. route and could confirm that only the i.n. route results in the production of IgA in the Bal and substantial numbers of antigen-specific T-cells in the lung. This is in line with other reports and supports the importance of the route of administration. Since substantial titers of neutralizing antibodies in sera were induced after vaccination at both locations, no differences in the protection against a SARS-CoV-2 infection could be observed in mice. In both cases, sterile immunity is postulated and there are only minor differences in the numbers of infiltrating cells in the lungs after the challenge (Fig.7). Most interestingly, there is still impaired replication and partial protection seen in animals which have received the adjuvant i.n, but not s.c. Unfortunately, this effect is not further evaluated and makes it difficult to conclude on the real contribution of the local antigen-specific immunity induced by spike-Pam₂cys. It would be interesting to see, what are the main effectors behind this antigen-unrelated protection? Are there increased levels of cytokines? Is there still an on-going inflammatory response three weeks after the treatment?

We thank the reviewer for these insightful comments. We agree that the effect of i.n adjuvant alone on protection from challenge at 3 weeks after vaccination is very interesting. To explore this further, we now report data on persistence of protection from infection in the revised manuscript, demonstrating sterilising lung immunity is maintained in vaccinated mice challenged with SARS-CoV-2 at 6 weeks post vaccination (Fig. 10). By contrast, the non-specific beneficial effect of i.n Pam₂Cys is lost at 6 weeks. Therefore, we now show that antigen-specific immunity is required to provide longer term protection from infection.

We also report additional longer-term immunogenicity studies assessing Spike-specific immune responses systemically and locally in the lungs at 8 weeks post final vaccination. We include here assessment of vaccination with antigen only, and demonstrate that this is not sufficient to generate protective nAb and T-cell responses (new Figs. 2, 3e,f, 4).

In the second part of the study, the authors wanted to analyze the contribution of TLR-2 signaling on hematopoietic and somatic cells for innate and adaptive responses by using BM-chimeras. They vaccinated the different BM-chimeras and analyzed the CD45+ cellular composition in lung and BAL 24h after one immunization or three weeks after the last of three immunizations. Briefly, they report that the early neutrophil infiltration depends on TLR-2 signaling on epithelial cells and the efficient recruitment of CD4 T-cells depends on TLR-2 signaling on CD45+ cells.

Overall, a protein-based vaccine suitable for mucosal vaccinations would be highly desirable and in advantage over vector-based approaches because of the pre-existing immunity and the ability to be used as repetitive booster vaccine. The present study offers some interesting insights in this topic, but also lack some substantial controls to really understand the protective mechanisms.

We agree with the reviewer that protein-based vaccines for mucosal delivery avoid pre-existing immunity against vectors, allowing them to be used repetitively to boost immunity and protection. We emphasise that protective immune mechanisms are well understood for COVID-19, primarily levels of neutralising antibody responses have been well correlated with protection (Khoury et al, *Nat Med*, 2021, p1205). This has been comprehensively assessed in our revised manuscript, with nAb titres determined in pseudovirus assays (Fig. 1c) or against ancestral SARS-CoV-2 infection, using serum and BALF from C57BL/6 mice (Fig. 1d, new 2c), TLR2KO/WT on a Balb/c background (Fig. 7b) and K18-hACE2 mice (Fig. 8b). We also assessed nAb in C57BL/6 background mice at various time points after vaccination in serum and BALF comparing SARS-CoV-2 ancestral, Beta, Delta and Omicron variants (Fig. 8b, new S1). We also assessed mucosal T-cell responses at early and late stages after vaccination (Figs. 3,4), and comprehensively profiled the role of mucosal TLR2 in the responses to vaccination with Pam₂Cys+Spike (Figs. 5,6 and7).

Major points:

1) A non-adjuvanted spike vaccine would be beneficial to judge on the adjuvant effect by Pam₂Cys. From the data, it is very obvious that Pam₂Cys plays a substantial role in inducing inflammation at the mucosal surface and even provide non-antigen-specific protection, but if this also beneficial for providing long-lasting memory responses is not clear. It would be therefore also necessary to study protection at later time points after the immunization where no unspecific effects of the adjuvant will be present anymore. Overall, the immunization schedule with 3 applications within 6 weeks and an observation period of 3 weeks post boost are rather short

We agree with the reviewer that it is important to examine persistence of vaccine-induced protective responses. We have performed a number of new experiments that include comparison of vaccination with non-adjuvanted Spike, showing the essential requirement for adjuvant to develop optimum adaptive immune responses to SARS-CoV-2. Additional results in the revised manuscript include:

1. Titration of anti-Spike IgG, IgG1, IgG2c, and IgA at eight weeks post final vaccination, in serum and BALF. This has been added in the new Fig 2. We demonstrate that vaccination by either route induces substantial anti-Spike IgG, IgG1 and IgG2c titres in both serum and BALF, but mucosal vaccination induces significantly greater titres in the airways and exclusively anti-Spike IgA.
2. Assessment of T-cell responses in the lungs at eight weeks post final vaccination, including Spike-specific CD4⁺ and CD8⁺ T-cell cytokine responses (revised Fig. 3 e,f). We also examined induction of tissue resident memory T-cells in the lungs post vaccination and identified that only mucosal vaccination with Pam₂Cys+Spike induced T_{RM}, in particular CD4⁺ T_{RM} in the lungs (new Fig 4, and lines 224-235 of the revised manuscript).
3. We also assessed Spike-specific antibody and T-cell responses after s.c or i.n vaccination with Spike protein antigen alone versus adjuvanted antigen (Pam₂Cys+Spike). We demonstrate the requirement for adjuvanted antigen, given mucosally, to induce IgA and maximise other antibody and lung T-cell responses. This is included in the new Fig 2, revised Fig 3e,f, and the new Fig 4.
4. Persistence of protection from infection: Protection from SARS-CoV-2 was examined in K18-hACE2 mice challenged at 3 or 6 weeks after final vaccination. At the later 6 week time, the innate resistance to infection provided by i.n Pam₂Cys alone was lost, whereas sterilising immunity after Pam₂Cys+Spike vaccination was maintained (see new Figure 10, and lines 516-528 of the revised manuscript). This highlights that while innate immune activation locally at the respiratory mucosa may provide some short-term benefit to resist viral infection, antigen-specific immunity is required to maintain protection.

2) The work on the BM-chimera is really time-intensive and laborious, but the results presented here offers only limited insights in the mechanisms and the real contributions of different TLR-2 expressing cell populations. First, there are only this two selected time points (early and late) and second a non-vaccinated or even better a non-adjuvanted control is missing. A non-vaccinated control would be needed to confirm no general differences in the homeostasis of the different CD45⁺ populations by the missing TLR-2 signal on

either the CD45+ or the epithelial compartment. Furthermore, this results are highly descriptive without direct information on the contribution to later observed immune responses. Concretely, is the early influx of neutrophils driven by TLR-2 signaling important for the IgG or IgA or CD4 T-cell response later on? To really generate mechanistic links would be very hard, but at least a kind of kinetic following the early events of cell migration and cytokine expression patterns would be nice to see.

While we agree with the reviewer that it may be of interest to examine kinetics of cell migration/cytokines in the bone marrow chimeras, we believe this is out of scope of the current manuscript. This work would take >6 months to generate the chimeras, and we do not consider this is ethically justified as it would provide limited additional information. The results provided are intended to be comparative, but the contribution to later immune responses can be inferred – neutrophil influx is not necessarily needed to drive the later adaptive response, rather TLR2 signalling on hematopoietic cells is required. In particular, we note that the capacity to respond to adjuvant via TLR2 is essential for nAb induction (Fig. 7b). We also note a previous published report profiling similar TLR2/WT chimeras in a naïve state, and this work is already referenced in the manuscript (Deliyannis et al, *JCI Insight*, 2021).

3) As mentioned in the summary, it would be highly informative to study the non-specific protection of Pam₂Cys alone and also to somehow establish correlates of protection for the vaccine. Since the neutralizing antibody response is so high in both vaccine groups that sterile immunity can be reached, it would be nice to see if the IgA response would be beneficial for neutralization at the mucosal surface, e.g. by neutralization assay with BAL samples against VOCs?

We note that other groups have described early non-specific protection of mucosal Pam₂Cys alone in animal models of IAV and SARS-CoV-2, as referenced in the Discussion (e.g refs 33, 34, 35, 48). In our revised manuscript we also include additional data on the failure of non-specific protection provided by i.n Pam₂Cys alone to persist. We also compare protection from SARS-CoV-2 in K18-hACE2 mice challenged at either 3 or 6 weeks after final vaccination. At the 6 week time point, the innate resistance to infection provided by i.n Pam₂Cys alone was lost, whereas sterilising immunity after Pam₂Cys+Spike vaccination was maintained (see new Figure 10 and lines 516-528 of the revised manuscript). This highlights that while innate immune activation locally at the respiratory mucosa may provide some short-term resistance to viral infection, antigen-specific immunity is required to maintain protection.

We emphasise that protective immune mechanisms are well understood for COVID-19, primarily levels of neutralising antibody responses have been well correlated with protection (Khoury et al, *Nat Med*, 2021, p1205), as included in our Discussion at lines 570-578 of the revised manuscript. We include additional data in this revision testing serum and BALF from one and 8-weeks post final vaccination to determine nAb titres against ancestral SARS-CoV-2, as well as VOCs Delta and Omicron BA.5 (new Fig. 2c, new Fig. S1). We have also added to the Discussion a recent publication detailing the differences in localised airway responses from individuals receiving peripheral vaccination versus SARS-CoV-2 infection, at lines 555-559, (new ref 36).

4) It is interesting to see that the T-cell response is dominated by Th17 CD4 T-cells and almost no TH1 CD4 or CD8 T-cells are induced. I think, it was expected from the previous TB study, but it would be interesting to see how these responses could contribute to the protection against SARS-CoV-2. This could be at least discusses later on.

It is not unusual for mucosal vaccines, particularly subunit vaccines with a variety of adjuvants, to induce a dominant Th17 CD4⁺ T-cell response, as discussed on lines 615-617 of the manuscript. However, we note the known plasticity of T-cell responses in vivo. We also highlight that new data in this revision shows the induction of a CD8⁺ lung T-cell response at 8 weeks post vaccination (new Fig. 3f) and induction of both CD4⁺ and CD8⁺ lung T_{RM} (new Fig 4).

Minor points:

5) Although I understand the value of unbiased TSNE-analyses of flow data, here it does not seem to be the best way to present the efficacy of bone marrow reconstitution (Fig. 3+4). From the plots in 3b and 4b, it is not obvious to which content the reconstitution was established and which cells mainly express TLR-2. For

that, one have to look in the complicated cluster figures in the supplement. It would be nice to show the TLR-2 expression on selected populations (similar to sup.4 or 6), and estimate the percentage of TLR2+ and TLR2- cells, e.g. CD4+ TLR2+/- or DC TLR2+/- . Only from that, one could judge on the efficient depletion of CD45+ cells by irradiation and exclude residual TLR2+ cells in WT-KO mice.

The unbiased nature of the FItSNE analyses of total CD45⁺ lung populations in our chimeric mouse models (Figs. 5,6) provides a global visualisation of TLR2 expression to assess effectiveness of hematopoietic depletion (by irradiation) and replacement (by bone marrow transfer). This form of visualisation for cytometry data is well accepted in the field. That being said, we have added annotation to Fig. 5b and 6b to make clear the host recipient phenotype and bone marrow phenotype. In addition, as requested by the reviewer, the phenotype of individual cell populations that most highly express TLR2, is included in the supplementary data in Figs. S6, S8 and S11b. We have added gates to these flow cytometry plots to indicate the % of TLR2⁺ cells in each of these key populations, to aid the reader. We have also added minor changes to the Results to clarify this (lines 289-292 of the revised manuscript).

6) The differences between the cell migration pattern in lung and BAL described for the BM-chimera are very interesting, but there is not much written or discussed about the potential reasons (e.g. different kinetics)

We highlight the effect of TLR2 activation on cell migration in the mucosal environment in the Discussion, with explanation of our data, at lines 601-617. After TLR2 stimulation, IL-6, TNF and IL-1 may be released to recruit neutrophils and activate APCs. We have added additional data in this revision, performing Legendplex assays in BALF from *Tlr2*^{-/-} and WT chimeric mice at 24 hours post one mucosal vaccination with Pam₂Cys+Spike. There was increased release of cytokines TNF and IL-6 only in the airways of mice with wild type hematopoietic cells, consistent with TLR2-mediated activation and recruitment of alveolar macrophages and monocytes to the BAL, as shown in Fig. S8. These cytokine data are included in new Fig. S9 and described in the text of the revised manuscript at lines 341-343.

Reviewer #3:

This is an interesting and informative set of experiments, presented in a scholarly fashion. For the most part, it explores the cell types that respond to the TL2 agonist, introduced intranasally. Both early innate and later adaptive and cellular recruitment to the respiratory tract are evaluated. The experimental results themselves are comprehensive, well designed and analyzed in a sophisticated manner.

There are a several limitations in the analyses, discussed below.

However, a central criticism of this manuscript is that it is a composite of fundamental issues of TLR2 agonist Pam₂Cys mediated activation and recruitment of cells to the respiratory tract, what cells are involved in signalling and what events follow ligand recognition, in both the antigen specific and antigen non-specific compartments. The studies here are well designed and well characterized.

But the primary premise of the paper is that this vaccine system will ultimately be useful for protection from SARS-CoV2. It seems that this is an add-on feature of the study to make it more timely. IL17 is the major lung cytokine produced and this cytokine as well as many of the cells that are recruited are well known to primarily protect against bacterial and fungal infections. This cytokine can also lead to lung damage and has a significant risk for proinflammatory responses in humans. Thus, the likelihood of this SARS vaccine strategy seem low.

We thank the reviewer for the positive comments on our work. We respectfully disagree with the central criticism that the manuscript is a composite of related stories. Our manuscript presents, for the first time, a novel mucosal vaccine for SARS-CoV-2 with convincing efficacy data in a murine model of virulent SARS-CoV-2 infection. This work is further expanded by mechanistic studies examining, in detail, immunological responses after mucosal vaccination with Pam₂Cys+Spike, including the role of TLR2 expression on respiratory epithelia and immune cells in the generation of immune responses against SARS-CoV-2 following mucosal vaccination. Examination of this approach for COVID-19 is not an add-on feature, but a valid

exploration based on prior data (Hanna and Ashhurst et al. PNAS 2021) indicating this mucosal vaccination technology can be highly effective at inducing antibody and T-cell responses in the airways.

With regard to the immune responses induced and the likelihood of this as a vaccine strategy for COVID-19:

1. We present lung histology of our vaccinated mice, post SARS-CoV-2 challenge, and show no damaging lung pathology in any of our vaccinated mice (Fig. 9).
2. In humans, it is likely that the vaccine would be used for intranasal delivery only, and so it is highly unlikely that such an approach would hold a significant risk for proinflammatory lung responses. Indeed, there are now multiple mucosal vaccines for COVID-19 in clinical trials, however only one other is a subunit vaccine and is given without an adjuvant. We demonstrate in this revision that Spike antigen without Pam₂Cys adjuvant was not as effective (see new Fig. 2).
3. It is important to note that IL-17 responses do not always lead to lung damage. Dominant Th17 responses are common following mucosal vaccination. Indeed our group has demonstrated in mucosal TB vaccine studies with other adjuvants (e.g Advax) that predominantly lung IL-17 responses were induced and these were essential for protection without increasing lung damage (Counoupas et al, *npj Vaccines*, 2020).

The authors observed protection from SARS in an B6 animal model, but these data are not compelling. First, the authors use an older transgenic model for ACE-2 expression that does not recapitulate normal expression of this SARS receptor in humans. Second, because this transgenic model is in B6 mice, and the TLR2 bone marrow chimera is developed and explored in BALB/c mice, it is difficult to extrapolate the first half of the paper with the second half. These two strains are dramatically different in T cell phenotype. So the events discovered in one strain cannot be expected to be replicated in another.

We agree with the reviewer that the K18-hACE2 model does not recapitulate normal expression of ACE2 in humans, as the hACE2 expression is greater and more widely distributed. Nevertheless, this provides a model of severe SARS-CoV-2-induced lung disease, so that generating protection in K18-hACE2 mice is a powerful indicator of efficacy and this continues to be a well-accepted model for vaccine development (Johansen et al, *Mucosal Immunol*, 2020).

Mucosal vaccination with Pam₂Cys/Spike generated sterilising immunity at 3 weeks post vaccination (Fig. 8), with no evidence of clinical disease or immunopathology (Fig. 8, 9). We also provide new data in this revision demonstrating that sterilising immunity is maintained at 6 weeks post vaccination (new Fig. 10). This is associated with strong nAb and T-cell responses specific to SARS-CoV-2. Our extensive data therefore fully supports this approach as a valid strategy for preventing SARS-CoV-2 infection and thus reducing ongoing transmission.

While we recognise that C57BL/6 and Balb/c mice have some differences in immune responses, there is clear evidence that the key mechanism of immunological protection from SARS-CoV-2 is neutralising antibody level (Khoury et al, *Nat Med*, 2021) that enables data to be compared between the two strains. We have demonstrated strong SARS-CoV-2-specific nAb responses in the two strains (Fig. 1c, Fig. 2c, 7b), as well as comparable Spike-specific T-cell cytokine responses in the strains (Fig. 3, Fig. 7d,e), and consider this a strength of the study. We also highlight that nAb responses in serum or BALF are only generated in settings where vaccinated mice can respond to the adjuvant signalling via TLR2 (Fig. 7b). The hACE2 transgenics were not available on a BALB/c background so we were unable to perform protection studies in this setting. Further, we do not consider that this would be ethically justified given the immunological data already provided.

Also, and most fundamentally, the events following TLR2 engagement and robust IL17 responses are infiltration of cells that may be useful for bacterial infections but unlikely to be optimal for a viral infection.

As described above there is clear evidence that the key mechanism of immunological protection from SARS-CoV-2 is neutralising antibody level (Khoury et al, *Nat Med*, 2021). In this revision, we have comprehensively demonstrated this nAb response (Fig. 1c, 2c, 7b, new S1), and Spike-specific T-cell cytokine responses in the lungs (Fig. 3, Fig. 7d,e) including generation of lung T_{RM} following mucosal vaccination (Fig. 4). Additionally,

vaccination with Pam₂Cys+Spike generated sterilising immunity at 3 weeks post vaccination (Fig. 8), with no evidence of clinical disease or immunopathology (Fig. 8, 9). We also provide new data demonstrating that this sterilising immunity is maintained at 6 weeks post vaccination (new Fig. 10). This is associated with strong nAb and T-cell responses specific to SARS-CoV-2. This fully supports this approach as a valid strategy for preventing SARS-CoV-2 infection.

Finally, a series of 3 sequential IN prime/boost vaccine regimens is hard to translate to a path in human SARS vaccine design

This is a first report and proof of principle study of this vaccine strategy for SARS-CoV-2 in animal models. Subsequent studies will examine prime/boost regimens including with reduced numbers of doses and mucosal boost post-parenteral prime. We also highlight that there are now multiple mucosal vaccines for COVID-19 in clinical trials, however only one other is a subunit protein vaccine and is given without an adjuvant. There is substantial clinical interest in this approach and we anticipate a clear pathway to translate nasal vaccines to humans as a booster following parenteral prime. The next steps for translating this vaccine strategy are discussed on lines 641-672 of the revised manuscript.

Other issues:

The T cell phenotype is not fully explored and there is inadequate treatment of cells producing multiple cytokines. The sources of IL-17 are only discussed in the context of T cells -the authors should determine sources of IL-17 by methods such as direct ex vivo staining.

In this revision we have added additional data to define the T-cell phenotype as requested by the reviewer:

1. Assessment of T-cell responses in the lungs at eight weeks post final vaccination, including Spike-specific CD4⁺ and CD8⁺ T-cell cytokine responses (revised Fig. 3 e,f). While we agree that in some contexts presenting the gating for multiple cytokine-producing T-cells is important, as well as non-T-cell sources of IL-17, in this study where antibody responses are of primary importance we believe this would be of limited value. Therefore, for the sake of clarity for the reader, we have maintained presentation of the T-cell data as proportion positive for each individual cytokine.
2. Induction of tissue resident memory T-cells in the lungs post vaccination and demonstration that only mucosal vaccination with Pam₂Cys+Spike induced T_{RM}, in particular for CD4⁺ T_{RM} in the lungs (new Fig 4, described in the text of the revised manuscript on lines 224-235).
3. Spike-specific T-cell responses after s.c or i.n vaccination with antigen alone (i.e Spike) versus adjuvanted antigen (Pam₂Cys+Spike). We demonstrate the requirement for adjuvanted antigen, given mucosally, to induce lung T-cell responses. This is included in revised Fig 3e,f, and new Fig 4.

Reviewers' Comments:

Reviewer #1:

Remarks to the Author:

The new data included in the revision directly address the main comments in the previous critique regarding longer term immunity, antibody isotypes, and so on. As noted previously this report will be of great interest, and adds significantly to the discussion of routes of immunization, adjuvants, as well as the question of whether mucosal IgA makes a significant impact on protective immunity.

Reviewer #2:

Remarks to the Author:

In this revised manuscript, the authors included new data on long-term immunogenicity and efficacy, which emphasizes the protection against SARS-CoV-2 infection is mediated by antigen-specific immunity. Furthermore, the adjuvant effect on the antigen-specific immunity is demonstrated by addition of the antigen only control. With these new data, my main concerns were adequately addressed and the conclusions are now very well supported by the experimental data.

Reviewer #3:

Remarks to the Author:

My concerns about the limitation of this study design-the multiple vaccinations and naive muring model system and unlikely path as a human vaccine for respiratory viruses persist. The concern regarding a Th17 dominated responses also has not been alleviated by the authors comments. Naive animals vs immune responses in humans, with pre-existing and complex immunity in the respiratory tract or that would be recalled post intranasal vaccination is a serious concern.

Nature Communications NCOMMS-21-50205A: Final Revisions

Ashhurst AS, et al; Mucosal TLR2-activating protein-based vaccination induces potent pulmonary immunity and protection against SARS-CoV-2 in mice.

Response to Reviewers' comments

Reviewer #1:

The new data included in the revision directly address the main comments in the previous critique regarding longer term immunity, antibody isotypes, and so on. As noted previously this report will be of great interest and adds significantly to the discussion of routes of immunization, adjuvants, as well as the question of whether mucosal IgA makes a significant impact on protective immunity.

We thank the reviewer. No changes requested.

Reviewer #2:

In this revised manuscript, the authors included new data on long-term immunogenicity and efficacy, which emphasizes the protection against SARS-CoV-2 infection is mediated by antigen-specific immunity. Furthermore, the adjuvant effect on the antigen-specific immunity is demonstrated by addition of the antigen only control. With these new data, my main concerns were adequately addressed, and the conclusions are now very well supported by the experimental data.

We thank the reviewer. No changes requested.

Reviewer #3:

My concerns about the limitation of this study design-the multiple vaccinations and naive muring model system and unlikely path as a human vaccine for respiratory viruses persist. The concern regarding a Th17 dominated responses also has not been alleviated by the authors comments. Naive animals vs immune responses in humans, with pre-existing and complex immunity in the respiratory tract or that would be recalled post intranasal vaccination is a serious concern.

The reviewer does not request any additional experiments. The comments are general in nature, and we make the following responses and have added a sentence to the Discussion to address these.

1. The murine SARS-CoV-2 infection model has been used in the development of the current COVID-19 vaccines as well as for other respiratory infections, and this is an appropriate model. Multiple doses are used for most current COVID-19 vaccines, as well as other vaccines, and is not necessarily a barrier to future mucosal vaccines against SARS-CoV-2. No changes to text.
2. The reviewer is concerned about the “Th17 dominated immune responses” to the vaccine. We consider it more appropriate to view this vaccine as inducing a “balanced” immune response as there is a very strong IgG and IgA antibody response consistent with T follicular helper cell responses in the lung draining lymph nodes (shown in Fig 7c) as well as Th17 responses. Moreover, mucosal vaccines delivered to the lung for other respiratory infections also induce a Th17-associated T cell response without lung damage (Ref 47: Counoupas et al, npj Vaccines, **5**, 105, 2020). We have added a sentence to the Discussion to state that:
Line 579: This lung Th17 response was balanced with a Tfh response in the draining lymph nodes (Fig 7c) and was not associated with evidence of inflammatory damage to the lungs following SARS-CoV-2 infection (Fig 9).
3. The reviewer is concerned that use of mucosal vaccines to SARS-CoV-2 in humans with pre-existing immunity to the virus could be a “serious concern” presumably causing

immunopathology in the respiratory tract. We acknowledge that pre-existing immunity in the respiratory tract in humans may impact responses to intranasal vaccination. This would need to be monitored in clinical trials and is therefore outside the scope of this manuscript. This concern would apply to any nasal or respiratory vaccine for SARS-CoV-2 or other respiratory infections and has not prevented clinical trials starting for more than eight mucosal COVID-19 vaccines, including one booster vaccine in subjects with pre-existing immunity. This is already mentioned in the Discussion (Lines: 601- 611).